# LEARNING FOR EDGE-WEIGHTED ONLINE BIPARTITE MATCHING WITH ROBUSTNESS GUARANTEES

## ABSTRACT

Many real-world problems, such as online ad display, can be formulated as online bipartite matching. The crucial challenge lies in the nature of sequentially-revealed online item information, based on which we make irreversible matching decisions at each step. While numerous expert online algorithms have been proposed with bounded worst-case competitive ratios, they may not offer satisfactory performance in average cases. On the other hand, reinforcement learning (RL) has been applied to improve the average performance, but they lack robustness and can perform arbitrarily badly. In this paper, we propose a novel RL-based approach to edge-weighted online bipartite matching with robustness guarantees (`LOMAR`), achieving both good average-case and good worst-case performance. The key novelty of `LOMAR` is a new online switching operation which, based on a judiciously-designed condition to hedge against future uncertainties, decides whether to follow the expert's decision or the RL decision for each online item arrival. We prove that for any $\rho \in [0, 1]$, `LOMAR` is $\rho$-competitive against any given expert online algorithm. To improve the average performance, we train the RL policy by explicitly considering the online switching operation. Finally, we run empirical experiments to demonstrate the advantages of `LOMAR` compared to existing baselines.

## 1 INTRODUCTION

Online bipartite matching is a classic online problem of practical importance (Mehta, 2013; Kim & Moon, 2020; Fahrbach et al., 2020; Antoniadis et al., 2020b; Huang & Shu, 2021; Gupta & Roughgarden, 2020). In a nutshell, online bipartite matching assigns online items to offline items in two separate sets: when an online item arrives, we need to match it to an offline item given applicable constraints (e.g., capacity constraint), with the goal of maximizing the total rewards collected (Mehta, 2013). For example, numerous applications, including scheduling tasks to servers, displaying advertisements to online users, recommending articles/movies/products, among many others, can all be modeled as online bipartite matching or its variants.

The practical importance, along with substantial algorithmic challenges, of online bipartite matching has received extensive attention in the last few decades (Karp et al., 1990; Fahrbach et al., 2020). Concretely, many algorithms have been proposed and studied for various settings of online bipartite matching, ranging from simple yet effective greedy algorithms to sophisticated ranking-based algorithms (Karp et al., 1990; Kim & Moon, 2020; Fahrbach et al., 2020; Aggarwal et al., 2011; Devanur et al., 2013). These expert algorithms typically have robustness guarantees in terms of the competitive ratio — the ratio of the total reward obtained by an online algorithm to the reward of another baseline algorithm (commonly the optimal offline algorithm) — even under adversarial settings given arbitrarily bad problem inputs (Karp et al., 1990; Huang & Shu, 2021). In some settings, even the optimal competitive ratio for adversarial inputs has been derived (readers are referred to (Mehta, 2013) for an excellent tutorial). The abundance of competitive online algorithms has clearly demonstrated the importance of performance robustness in terms of the competitive ratio, especially in safety-sensitive applications such as matching mission-critical items or under contractual obligations (Fahrbach et al., 2020). Nonetheless, as commonly known in the literature, the necessity of conservativeness to address the worst-case adversarial input means that the average performance is typically not optimal (see, e.g., (Christianson et al., 2022; Zeynali et al., 2021) for discussions in other general online problems).

More recently, online optimizers based on reinforcement learning (RL) (Chen et al., 2022; Georgiev & Lió, 2020; Wang et al., 2019; Alomrani et al., 2021; Du et al., 2019; Zuzic et al., 2020) have been proposed in the context of online bipartite matching as well as other online problems. Specifically, by exploiting statistical information of problem inputs, RL models are trained offline and then applied online to produce decisions given unseen problem inputs. These RL-based optimizers can often achieve high average rewards in many typical cases. Nonetheless, they may not have any performance robustness guarantees in terms of the competitive ratio. In fact, a crucial pain point is that the worst-case performance of many RL-based optimizers can be arbitrarily bad, due to, e.g., testing distribution shifts, inevitable model generalization errors, finite samples, and/or even adversarial inputs. Consequently, the lack of robustness guarantees has become a key roadblock for wide deployment of RL-based optimizers in real-world applications.

In this paper, we focus on an important and novel objective — achieving both good average performance and guaranteed worst-case robustness — for *edge-weighted* online bipartite matching (Fahrbach et al., 2020; Kim & Moon, 2020). More specifically, our algorithm, called LOMAR (Learning-based approach to edge-weighted Online bipartite MAtching with Robustness guarantees), integrates an expert algorithm with RL. The key novelty of LOMAR lies in a carefully-designed online *switching* step that dynamically switches between the RL decision and the expert decision online, as well as a switching-aware training algorithm. For both no-free-disposal and free-disposal settings, we design novel switching conditions as to when the RL decisions can be safely followed while still guaranteeing robustness of being $\rho$-competitive against *any* given expert online algorithms for any $\rho \in [0, 1]$. Furthermore, if the expert itself has a competitive ratio of $\lambda \leq 1$ against the optimal offline algorithm (OPT), then it will naturally translate into LOMAR being $\rho\lambda$-competitive against OPT. To improve the average performance of LOMAR, we train the RL policy in LOMAR by explicitly taking into account the introduced switching operation. Importantly, to avoid the "no supervision" trap during the initial RL policy training, we propose to approximate the switching operation probabilistically. Finally, we offer empirical experiments to demonstrate that LOMAR can improve the average cost (compared to existing expert algorithms) as well as lower the competitive ratio (compared to pure RL-based optimizers).

## 2 RELATED WORKS

Online bipartite matching has been traditionally approached by expert algorithms (Mehta, 2013; Karande et al., 2011; Huang et al., 2019; Devanur et al., 2013). A simple but widely-used algorithm is the (deterministic) greedy algorithm (Mehta, 2013), achieving reasonably-good competitive ratios and empirical performance (Alomrani et al., 2021). Randomized algorithms have also been proposed to improve the competitive ratio (Ting & Xiang, 2014; Aggarwal et al., 2011). In addition, competitive algorithms based on the primal-dual framework have also been proposed (Mehta, 2013; Buchbinder et al., 2009). More recently, multi-phase information and predictions have been leveraged to exploit stochasticity within each problem instance and improve the algorithm performance (Kesselheim et al., 2013). For example, (Korula & Pál, 2009) designs a secretary matching algorithm based on a threshold obtained using the information of phase one, and exploits the threshold for matching in phase two. Note that stochastic settingsconsidered by expert algorithms (Mehta, 2013; Karande et al., 2011) mean that the arrival orders and/or rewards of different online items within each problem instance are stochastic. By contrast, as shown in equation 2, we focus on an unknown distribution of problem instances whereas the inputs within each instance can still be arbitrary.

Another line of algorithms utilize RL to improve the average performance (Wang et al., 2019; Georgiev & Lió, 2020; Chen et al., 2022; Alomrani et al., 2021). Even though heuristic methods (such as using adversarial training samples (Zuzic et al., 2020; Du et al., 2022)) are used to empirically improve the robustness, they do not provide any theoretically-proved robustness guarantees.

ML-augmented algorithms have been recently considered for various problems (Rutten et al., 2022; Christianson et al., 2022; Chłędowski et al., 2021; Lykouris & Vassilvitskii, 2021; Gupta & Roughgarden, 2017). By viewing the ML prediction as blackbox advice, these algorithms strive to provide good competitive ratios when the ML predictions are nearly perfect, and also bounded competitive ratios when ML predictions are bad. But, they still focus on the worst case without addressing the average performance or how the ML model is trained. By contrast, the RL model in LOMAR is trained by taking into account the switching operation and performs inference based on the actual state

(rather than its own independently-maintained state as a blackbox). Assuming a given downstream algorithm, (Wang et al., 2021; Liu & Grigas, 2021; Wilder et al., 2019; Elmachtoub & Grigas, 2017; Du et al., 2021; Anand et al., 2021) focus on learning the ML model to better serve the end goal in completely different (sometimes, offline optimization) problems.

LOMAR is relevant to conservative bandits/RL (Wu et al., 2016; Kazerouni et al., 2017; Yang et al., 2022; Garcelon et al., 2020). With unknown reward functions (as well as transition models if applicable), conservative bandits/RL leverages an existing policy to safeguard the exploration process. But, they only consider the cumulative reward without addressing future uncertainties when deciding exploration vs. rolling back to an existing policy. Thus, as shown in Section 4, this cannot guarantee robustness in our problem. Also, constrained policy optimization (Yang et al., 2020; Kumar et al., 2020; Schulman et al., 2015; Achiam et al., 2017; Thomas et al., 2021; Berkenkamp et al., 2017) focuses on average (cost) constraints in the long run, whereas LOMAR achieves stronger robustness (relative to an expert algorithm) for each episode with even adversarial inputs.

## 3 PROBLEM FORMULATION

We focus on *edge-weighted* online bipartite matching, which includes un-weighted and vertex-weighted matching as special cases (Fahrbach et al., 2020; Kim & Moon, 2020). In the following, we also drop "edge-weighted" if applicable when referring to our problem.

The goal of the agent is to match items (a.k.a. vertices) between two sets $\mathcal{U}$ and $\mathcal{V}$ to gain as high total rewards as possible. Suppose that $\mathcal{U}$ is fixed and contains *offline* items $u \in \mathcal{U}$, and that the *online* items $v \in \mathcal{V}$ arrive sequentially: in each time slot, an online item $v \in \mathcal{V}$ arrives and the weight/reward information $\{w_{uv} \mid w_{u,\min} \leq w_{uv} \leq w_{u,\max}, u \in \mathcal{U}\}$ is revealed, where $w_{uv}$ represents the reward when the online item $v$ is matched to each offline $u \in \mathcal{U}$. We denote one problem instance by $\mathcal{G} = \{\mathcal{U}, \mathcal{V}, \mathcal{W}\}$, where $\mathcal{W} = \{w_{uv} \mid u \in \mathcal{U}, v \in \mathcal{V}\}$. We denote $x_{uv} \in \{0, 1\}$ as the matching decision indicating whether $u$ is matched to $v$. Also, any offline item $u \in \mathcal{U}$ can be matched up to $c_u$ times, where $c_u$ is essentially the capacity for offline item $u$ known to the agent. The objective is to maximize the total collected reward $\sum_{v \in \mathcal{V}, u \in \mathcal{U}} x_{uv} w_{uv}$. With a slight abuse of notations, we denote $x_v \in \mathcal{U}$ as the index of item in $\mathcal{U}$ that is matched to item $v \in \mathcal{V}$. The set of online items matched to $u \in \mathcal{U}$ is denoted as $\mathcal{V}_u = \{v \in \mathcal{V} \mid x_{uv} = 1\}$.

The edge-weighted online bipartite matching problem has been mostly studied under two different settings: no free disposal and with free disposal (Mehta, 2013). In the no-free-disposal case, each offline item $u \in \mathcal{U}$ can only be matched strictly up to $c_u$ times; in the free-disposal case, each offline item $u \in \mathcal{U}$ can be matched more than $c_u$ times, but only the top $c_u$ rewards are counted when more than $c_u$ online items are matched to $u$. Compared to the free-disposal case, the no-free-disposal case is significantly more challenging with the optimal competitive ratio being 0 in the strong adversarial setting unless additional assumptions are made (e.g., $w_{u,\min} > 0$ for each $u \in \mathcal{U}$ (Kim & Moon, 2020) and/or random-order of online arrivals) (Fahrbach et al., 2020; Mehta, 2013). The free-disposal setting not only makes the problem more tractable, but also is practically motivated by the display ad application where the advertisers (i.e., offline items $u \in \mathcal{U}$) will not be unhappy if they receive more impressions (i.e., online items $v \in \mathcal{V}$) than their budgets $c_u$, even though only the top $c_u$ items count.

LOMAR can handle both no-free-disposal and free-disposal settings. For better presentation of our key novelty and page limits, we focus on the no-free-disposal setting in the body of the paper, *while deferring the free-disposal setting to Appendix B*. Specifically, the *offline* problem with no free disposal can be expressed as:

$$\max_{x_{uv} \in \{0,1\}, u \in \mathcal{U}, v \in \mathcal{V}} \sum x_{uv} w_{uv}, \text{ s.t., } \sum_{v \in \mathcal{V}} x_{uv} \leq c_u, \text{ and } \sum_{u \in \mathcal{U}} x_{uv} \leq 1, \forall u \in \mathcal{U}, v \in \mathcal{V}, \quad (1)$$

where the constraints specify the offline item capacity limit and each online item $v \in \mathcal{V}$ can only be matched up to one offline item $u \in \mathcal{U}$.

Given a problem instance $\mathcal{G}$ and an online algorithm $\alpha$, we use $f_u^\alpha(\mathcal{G})$ to denote the total reward collected for offline item $u \in \mathcal{U}$, and $R^\alpha(\mathcal{G}) = \sum_{u \in \mathcal{U}} f_u^\alpha(\mathcal{G})$ to denote the total collected reward. We will also drop the superscript $\alpha$ for notational convenience wherever applicable.

**Objective.** Solving the problem in equation 1 is very challenging in the online case, where the agent has to make irreversible decisions without knowing the future online item arrivals. Next, we first define the following generalized competitiveness as a metric of robustness.

**Definition 1** (Competitiveness). An online bipartite matching algorithm $\alpha$ is said to be $\rho-$competitive with $\rho \geq 0$ against the algorithm $\pi$ if for any problem instance $\mathcal{G}$, its total collected reward $R^\alpha(\mathcal{G})$ satisfies $R^\alpha(\mathcal{G}) \geq \rho R^\pi(\mathcal{G}) - B$, where $B \geq 0$ is a constant independent of the problem input, and $R^\pi$ is the total reward of the algorithm $\pi$.

Competitiveness against a given (online) algorithm $\pi$ is common in the literature (Christianson et al., 2022): the greater $\rho \geq 0$, the better robustness of the online algorithm, although the average rewards can be worse. Additionally, the constant $B \geq 0$ relaxes the strict competitive ratio by allowing an additive *regret* (Antoniadis et al., 2020a). When $B = 0$, the competitive ratio becomes the strict one.

In this paper, we focus on a setting where the problem instance $\mathcal{G} = \{\mathcal{U}, \mathcal{V}, \mathcal{W}\}$ follows an *unknown* distribution, whereas both the rewards $\mathcal{W}$ and online arrival order within each instance $\mathcal{G}$ can be adversarial. We consider both average performance and worst-case robustness guarantees as formalized below:

$$\max \mathbb{E}_\mathcal{G} \left[ R^\alpha(\mathcal{G}) \right], \quad \text{s.t. } R^\alpha(\mathcal{G}) \geq \rho R^\pi(\mathcal{G}) - B, \quad \forall \mathcal{G}, \tag{2}$$

where the expectation $\mathbb{E}_\mathcal{G} \left[ R^\alpha(\mathcal{G}) \right]$ is over the randomness $\mathcal{G} = \{\mathcal{U}, \mathcal{V}, \mathcal{W}\}$.

Note carefully that some manually-designed algorithms focus on a *stochastic* setting where the arrival order is random and/or the rewards $\{w_{uv} \mid w_{u,\min} \leq w_{uv} \leq w_{u,\max}, u \in \mathcal{U}\}$ of each online item is independently and identically distributed (i.i.d.) within each problem instance $\mathcal{G}$ Mehta (2013). By contrast, our settings are significantly different — we only assume an unknown distribution for the entire problem instance $\mathcal{G} = \{\mathcal{U}, \mathcal{V}, \mathcal{W}\}$ while both the rewards $\mathcal{W}$ and online arrival order within each instance $\mathcal{G}$ can be adversarial in our problem.

## 4 ONLINE SWITCHING FOR ROBUSTNESS GUARANTEES

We present LOMAR, which includes an online *switching* operation to dynamically decide to follow the ML decision or the expert decision, to achieve robustness guarantees with respect to the expert.

### 4.1 ONLINE SWITCHING

While switching is common in online algorithms, "*how to switch*"is highly non-trivial and a key merit for algorithm designs Antoniadis et al. (2020a); Christianson et al. (2022); Rutten et al. (2022). To guarantee robustness (i.e., $\rho$-competitive against a given expert for any $\rho \in [0, 1]$), we propose a novel online algorithm (Algorithm 1). In the algorithm, we independently run an expert online algorithm $\pi$ — the cumulative reward and item matching decisions are all maintained virtually for the expert, but not used as the actual decisions. Meanwhile, instead of being independently executed to provide blackbox advice based on its own virtual state (like in the prior ML-augmented online algorithm (Christianson et al., 2022)), the RL model in LOMAR makes online decisions based on the actual state at each step.

The most crucial step for safeguarding RL decisions is online switching: Lines 13–19 in Algorithm 1. The key idea for this step is to switch between the expert decision $x_v^\pi$ and the RL decision $\tilde{x}_v$ in order to ensure that the actual online decision $x_v$ meets the $\rho$-competitive requirement (against the expert $\pi$). Specifically, we follow the RL decision $\tilde{x}_v$ only if it can safely hedge against any future uncertainties (i.e., the expert's future reward increase); otherwise, we need to roll back to the expert's decision $x_v^\pi$ to stay on track for robustness.

Note that naive switching conditions, e.g., only ensuring that the actual cumulative reward is at least $\rho$ times of the expert's cumulative reward at each step (Wu et al., 2016; Yang et al., 2022), can fail to meet the competitive ratio requirement in the end. The reason is that, even though the competitive ratio requirement is met (i.e., $R_v \geq \rho R_v^\pi - B$) at the current step $v$, the expert can possibly obtain much higher rewards from future online items $v + 1, v + 2, \cdots$ if it has additional offline item capacity that the actual algorithm LOMAR does not have. Thus, we must carefully design the switching conditions to hedge against future risks. The no-free-disposal and free-disposal settings

---

**Algorithm 1** Inference of Robust Learning-based Online Bipartite Matching (LOMAR)

---

1: **Initialization:** The actual set of items matched to $u \in \mathcal{U}$ is $\mathcal{V}_{u,v}$ after sequentially-arriving item $v$'s assignment with $\mathcal{V}_{u,0} = \emptyset$, the actual remaining capacity is $b_u = c_u$ for $u \in \mathcal{U}$, and the actual cumulative reward is $R_0 = \sum_{u \in \mathcal{U}} f_u(\mathcal{V}_{u,0}) = 0$. The same notations apply to the expert algorithm $\pi$ by adding the superscript $\pi$. Competitive ratio requirement $\rho \in [0,1]$ and slackness $B \geq 0$ with respect to the expert algorithm $\pi$.
2: **for** $v = 1$ to $|\mathcal{V}|$ **do**
3:     Run the algorithm $\pi$ and match the item $v$ to $u \in \mathcal{U}$ based on the expert's decision $u = x_v^\pi$.
4:     Update the expert's decision set and reward for offline item $u = x_v^\pi$:
    $\mathcal{V}_{x_v^\pi,v}^\pi = \mathcal{V}_{x_v^\pi,v-1}^\pi \bigcup \{v\}$ and $f_{x_v^\pi} = f_{x_v^\pi}(\mathcal{V}_{x_v^\pi,v}^\pi)$.
5:     Update the expert's cumulative reward $R_v^\pi = \sum_{u \in \mathcal{U}} f_u$
6:     Get the set of available items $\mathcal{U}_a = \{u \in \mathcal{U} \mid |\mathcal{V}_{u,v-1}| < c_u\}$
7:     **for** $u$ in $\mathcal{U}_a$ **do**
8:         Collect the available history information $I_u$ about item $u$
9:         Run the RL model to get score: $s_u = w_{uv} - h_\theta(I_u, w_{uv})$ where $\theta$ is the network weight
10:     **end for**
11:     Calculate the probability of choosing each available item or skip:
    $\{\{\tilde{s}_u\}_{u \in \mathcal{U}_a}, \tilde{s}_{\text{skip}}\} = \text{softmax}\{\{s_u\}_{u \in \mathcal{U}_a}, 0\}$.
12:     Obtain RL decision: $\tilde{x}_v = \arg\max_{u \in \mathcal{U}_a \bigcup \{\text{skip}\}} \{\{\tilde{s}_u\}_{u \in \mathcal{U}_a}, \tilde{s}_{\text{skip}}\}$.
13:     **if** $R_{v-1} + w_{\tilde{x}_v,v} \geq \rho \left( R_v^\pi + \sum_{u \in \mathcal{U}} \left(|\mathcal{V}_{u,v-1}| - |V_{u,v}^\pi| + \mathbb{I}_{u=\tilde{x}_v}\right)^+ \cdot w_{u,\max} \right) - B$ **then**
14:         Select $x_v = \tilde{x}_v$.   //Follow the RL decision
15:     **else if** $x_v^\pi$ is available for matching (i.e., $|\mathcal{V}_{x_v^\pi,v-1}| < c_{x_v^\pi}$) **then**
16:         Select $x_v = x_v^\pi$.   //Follow the expert
17:     **else**
18:         Select $x_v = \text{skip}$.
19:     **end if**
20:     Update assignment and reward: $\mathcal{V}_{x_v,v} = \mathcal{V}_{x_v,v-1} \bigcup \{v\}$ and $R_v = R_{v-1} + w_{x_v,v}$
21: **end for**

---

require different switching conditions. Due to the page limit, we focus on the no-free-disposal setting below, while referring readers to Appendix B for more details about the free-disposal setting.

## 4.2 ROBUSTNESS CONSTRAINT

In the no-free-disposal case, an offline item $u \in \mathcal{U}$ cannot receive any additional online items if it has been matched for $c_u$ times up to its capacity. By assigning more online items to $u \in \mathcal{U}$ than the expert algorithm at step $v$, LOMAR can possibly receive a higher cumulative reward than the expert's cumulative reward. But, such advantages are just *temporary*, because the expert may receive an even higher reward in the future by filling up the unused capacity of item $u$. Thus, to hedge against the future uncertainties, LOMAR chooses the RL decisions only when the following condition is satified:

$$R_{v-1} + w_{\tilde{x}_v,v} \geq \rho \left( R_v^\pi + \sum_{u \in \mathcal{U}} \left(|\mathcal{V}_{u,v-1}| - |V_{u,v}^\pi| + \mathbb{I}_{u=\tilde{x}_v}\right)^+ \cdot w_{u,\max} \right) - B, \tag{3}$$

where $\mathbb{I}_{u=\tilde{x}_v} = 1$ if and only if $u = \tilde{x}_v$ and 0 otherwise, $(\cdot)^+ = \max(\cdot, 0)$, $\rho \in [0,1]$ and $B \geq 0$ are the hyperparameters indicating the desired robustness with respect to the expert algorithm $\pi$. The interpretation of equation 3 is as follows. The left-hand side is the total reward of LOMAR after assigning the online item $v$ based on the RL decision (i.e. $\tilde{x}_t$). The right-hand side is the expert's cumulative cost $R_v^\pi$, plus the term $\sum_{u \in \mathcal{U}} \left(|\mathcal{V}_{u,v-1}| - |V_{u,v}^\pi| + \mathbb{I}_{u=\tilde{x}_v}\right)^+ \cdot w_{u,\max}$ which indicates the maximum reward that can be possibly received by the expert in the future. This reservation term is crucial, especially when the expert has more unused capacity than LOMAR. Specifically, $|\mathcal{V}_{u,v-1}|$ is the number of online items (after assigning $v-1$ items) already assigned to the offline item $u \in \mathcal{U}$, and hence $\left(|\mathcal{V}_{u,v-1}| - |V_{u,v}^\pi| + \mathbb{I}_{u=\tilde{x}_v}\right)^+$ represents the number of more online items that LOMAR has assigned to $u$ than the expert if LOMAR follows the RL decision at step $v$. If LOMAR assigns fewer items than the expert for an offline item $u \in \mathcal{U}$, there is no need for any hedging because LOMAR is guaranteed to receive more rewards by filling up the item $u$ up to the expert's assignment level.

The term $w_{u,\max}$ in equation 3 is the set as the maximum possible reward for each decision. Even when $w_{u,\max}$ is unknown in advance, LOMAR still applies by simply setting $w_{u,\max} = \infty$. In this case, LOMAR will be less "greedy" than the expert and never use more resources than the expert at any step for any $u \in \mathcal{U}$.

## 4.3 ROBUSTNESS ANALYSIS

We now formally show the competitive ratio of LOMAR. The proof is available in the appendix.

**Theorem 4.1.** *For any $0 \leq \rho \leq 1$ and $B \geq 0$ and any expert algorithm $\pi$, LOMAR achieves a competitive ratio of $\rho$ against the algorithm $\pi$, i.e., $R \geq \rho R^\pi - B$ for any problem input.*

The hyperparameters $0 \leq \rho \leq 1$ and $B \geq 0$ govern the level of robustness we would like to achieve, at the potential expense of average reward performance. For example, by setting $\rho = 1$ and $B = 0$, we achieve the same robustness as the expert but leave little to none freedom for RL decisions. On the other hand, by setting a small $\rho > 0$ and/or large $B$, we provide higher flexibility to RL decisions for better average performance, while potentially decreasing the robustness. In fact, designing an algorithm that is guaranteed to simultaneously outperform RL and the expert is very challenging, if not impossible, and such tradeoff is necessary in the broad context of ML-augmented online algorithms (Rutten et al., 2022; Christianson et al., 2022). Additionally, in case of multiple experts, we can first combine these experts into a single expert and then apply LOMAR as if it works with a single combined expert.

While the competitive ratio of all online algorithms against the optimal offline algorithm is zero in the no-free-disposal and general adversarial setting, there exist provably competitive online expert algorithms under some mild assumptions and other settings (Mehta, 2013). For example, the simple greedy algorithm achieves $\left(1 + \max_{u \in \mathcal{U}} \frac{w_{u,\max}}{w_{u,\min}}\right)^{-1}$ under bounded weights assumptions for the adversarial no-free-disposal setting (Kim & Moon, 2020), and $\frac{1}{2}$ for the free-disposal setting (Fahrbach et al., 2020), and there also exist $1/e$-competitive algorithms against the optimal offline algorithm for the random-order setting (Mehta, 2013). Thus, an immediate result follows.

**Corollary 4.1.1.** *For any $0 \leq \rho \leq 1$ and $B \geq 0$, by using Algorithm 1 and an expert online algorithm $\pi$ that is $\lambda$-competitive against the optimal offline algorithm OPT, then under the same assumptions for $\pi$ to be $\lambda$-competitive, LOMAR is $\rho\lambda$-competitive against OPT.*

Corollary 4.1.1 provides a general result that applies to any $\lambda$-competitive expert algorithm $\pi$ under its respective required assumptions. For example, if the expert $\pi$ assumes an adversarial or random-order setting, then Corollary 4.1.1 also holds true under the same adversarial or random-order setting.

Finally, we comment on the randomized setting where randomization is over the algorithm choice and potentially increases the competitive ratio (Gupta & Roughgarden, 2020; Mehta, 2013). For the randomized setting, the competitive ratio is modified as $\mathbb{E}(R) \geq \rho R^{OPT} - B$, where $\mathbb{E}$ is the expectation and $R^{OPT}$ is the optimal reward of the optimal offline algorithm (Mehta, 2013). We make no assumptions on the expert $\pi$ in Algorithm 1. Thus, if the expert $\pi$ itself is randomized in Algorithm 1, then LOMAR will also be randomized. Also, for any $\rho \in [0, 1]$ and $B \geq 0$, by directly applying Theorem 4.1 and Corollary 4.1.1 to the randomized setting, the competitive ratio of LOMAR will be $\rho\lambda$-competitive against OPT if the randomized expert $\pi$ itself is $\lambda$-competitive against OPT.

## 5 RL POLICY TRAINING WITH ONLINE SWITCHING

The existing ML-augmented online algorithms typically assume a pre-trained standalone RL model (Christianson et al., 2022; Rutten et al., 2022). While the standalone RL model may perform well on its own, some already good actions can be replaced by expert's action due to switching for robustness in inference. In other words, there will be a objective mismatch between training and testing. To rectify the mismatch, we propose a novel approach to train the RL model in LOMAR by explicitly considering the switching operation.

**RL architecture.** For solving online bipartite matching, there exist various network architectures, e.g., fully-connected networks and scalable invariant network for arbitrary graph sizes. The prior study (Mehta, 2013) has shown using extensive empirical experiments that the invariant network

architecture, where each offline-online item pair runs through a separate neural network with shared weights among all the item pairs, is empirically advantageous, due to its scalability to large graph sizes and high average performance. We denote the RL model as $h_\theta(I_u, w_{uv})$ where $\theta$ is the network parameter. By feeding the item weight $w_{uv}$ and applicable history information $I_u$ for each offline-online item pair $(u, v)$, we can use the RL model to output a *threshold* for possible item assignment, following threshold-based algorithms (Huang et al., 2019; Mehta, 2013). The history information $I_u$ includes, but is not limited to, the average value and variance of weights assigned to $u$, average in-degree of $u$, and maximum weight for the already matched items. More details about the information can be found in the appendix. Then, with the RL output, we can obtain a score $s_u = w_{uv} - h_\theta(I_u, w_{uv})$, for each possible assignment, and the RL uses the offline item $u \in \mathcal{U}$ (plus "skip" with $s_{skip} = 0$ in the no-free-disposal setting) with the largest $s_u$ as its candidate action $\tilde{x}_v$ when checking the switching condition in Algorithm 1.

**Policy training.** Training the RL model by considering switching in Algorithm 1 is highly non-trivial. Most critically, the initial RL decisions can perform arbitrarily badly upon policy initialization, which means that the initial RL decisions are almost always overridden by the expert's decisions for robustness. Due to following the expert's decisions, the RL agent almost always receive a good reward, which actually has nothing to do with the RL's own decisions and hence provides little to no supervision to improve the RL policy. Consequently, this creates a *gridlock* for RL policy training. While using an offline pre-trained standalone RL model without considering online switching (e.g., (Alomrani et al., 2021)) as an initial policy may partially address this gridlock, this is certainly inefficient as we have to spend resources for training another RL model, let alone the likelihood of being trapped into the standalone RL model's suboptimal policies (e.g. local minimums).

To address these issues, we introduce another softmax probability with temperature $t$ to approximate the hard switching process during training. The switching probability depends on the cumulative reward difference $R_{diff}$ in the switching condition, which is

$$R_{diff} = R_{v-1} + w_{\tilde{x}_v, v} - \rho \left( R_v^\pi + \sum_{u \in \mathcal{U}} \left( |\mathcal{V}_{u,v-1}| - |V_{u,v}^\pi| + \mathbb{I}_{u=\tilde{x}_v} \right)^+ \cdot w_{u,\max} \right) + B \quad (4)$$

Then the probability of following RL is $p_{os} = \frac{e^{R_{diff}/t}}{1+e^{R_{diff}/t}}$, where $t$ is the temperature of softmax function. This softmax probability is differentiable and hence allows backpropagation to supervise the training of the RL model weight $\theta$. We train the RL agent by applying REINFORCE (Williams, 1992) to optimize the policy parameter $\theta$. Denote $\tau = \{x_1, \cdots, x_v\}$ as an action trajectory sample and $p_\theta(\tau)$ as the possibility of the trajectory given the RL policy, where $p_\theta$ is calculated based on the selection probability of RL model and expert.

Our goal is to maximize the expected total reward $R_\theta = \mathbb{E}_{\tau \sim p_\theta}[w_{x_v, v}]$. Thus, at each training step, given an RL policy with parameter $\theta$, we sample $n$ action trajectories $\{\tau_i = \{x_{1,i}, \cdots, x_{v,i}\}, i \in [n]\}$ and record the corresponding rewards. We can get the approximated average reward as $\hat{R}_\theta = \frac{1}{n} \sum_{i=1}^n w_{x_{i,v}, v}^i$. Then, we calculate the gradient of the RL policy parameter as $\nabla_\theta \hat{R}_\theta = \sum_{i=1}^n \left( \sum_{v \in \mathcal{V}} \nabla_\theta \log p_\theta(x_{v,i} \mid I_{u,i}) \right) \left( \sum_{v \in \mathcal{V}} w_{x_{v,i}, v}^i \right)$. Then, we update the parameter $\theta$ by $\theta = \theta + \alpha \nabla_\theta \hat{R}_\theta$, where $\alpha$ controls the training step size.

By changing the temperature $t$ for softmax, we are able to balance exploration and exploitation. Specifically, at the beginning of the policy training, we can set a high temperature to encourage the RL model to explore more aggressively, instead of sticking to the expert's decisions. As the RL model performance continuously improves, we can reduce the temperature in order to make the RL agent more aware of the downstream switching operation. The training process is performed offline as in the existing RL-based optimizers (Alomrani et al., 2021; Du et al., 2022) and described in Algorithm 3.

## 6 EXPERIMENT

### 6.1 SETUP

To validate the effectiveness of LOMAR, we conduct experiments based on the movie recommendation application. Specifically, when an user (i.e., online item $v$) arrives, we recommend a movie (i.e.,

offline item $u$) to this user and receive a reward based on the user-movie preference information. We choose the MovieLens dataset (Harper & Konstan, 2015), which provides a total of 3952 movies, 6040 users and 100209 ratings. We preprocess the dataset to sample movies and users randomly from the dataset to generate subgraphs, following the same steps as used by (Dickerson et al., 2019) and (Alomrani et al., 2021). In testing dataset, we empirically evaluate each algorithm using average reward (AVG) and competitive ratio (CR, against OPT), which represents the average performance and worst case performance, respectively. Thus, the value of CR is the empirically worst reward ratio in the testing dataset. For fair comparison, all the experimental settings like capacity $c_u$ follow those used in (Alomrani et al., 2021). More details about the problem setup and training details are deferred to Appendix A.

**Baseline Algorithms.** We consider the following baselines. All the RL policies are trained offline with the same architecture and applicable hyperparameters.

- OPT: The offline optimal oracle has the complete information about the bipartite graph. We use the Gurobi optimizer to find the optimal offline solution.

- Greedy: At each step, Greedy selects the available offline item with highest weight.

- DRL: It uses the same architecture as in LOMAR, but does not consider online switching for training or inference. That is, the RL model is both trained and tested with $\rho = 0$. More specifically, our RL architecture has 3 fully connected layers, each with 100 hidden nodes.

- DRL-OS (DRL-OnlineSwitching): We apply online switching to the same RL policy used by DRL during inference. That is, the RL model is trained with $\rho = 0$, but tested with a different $\rho > 0$.

The choice of baselines include all those considered in (Alomrani et al., 2021). In the no-free-disposal setting, the best competitive ratio is 0 in general adversarial cases (Mehta, 2013). Here, we use Greedy as the expert algorithm, because the recent study (Alomrani et al., 2021) has shown that Greedy performs better than other alternatives and is a strong baseline.

## 6.2 RESULTS

| Test | DRL-OS | | LOMAR ($\rho = 0.4$) | | LOMAR ($\rho = 0.6$) | | LOMAR ($\rho = 0.8$) | | Greedy | |
|---|---|---|---|---|---|---|---|---|---|---|
| | AVG | CR | AVG | CR | AVG | CR | AVG | CR | AVG | CR |
| $\rho = 0.4$ | 12.315 | 0.800 | **12.364** | **0.819** | 12.288 | 0.804 | 12.284 | 0.804 | 11.000 | 0.723 |
| $\rho = 0.6$ | 11.919 | 0.787 | 11.982 | 0.807 | **11.990** | **0.807** | 11.989 | 0.800 | 11.000 | 0.723 |
| $\rho = 0.8$ | 11.524 | **0.773** | 11.538 | 0.766 | 11.543 | 0.762 | **11.561** | 0.765 | 11.000 | 0.723 |

Table 1: Comparison under different $\rho$. In the top, LOMAR ($\rho = x$) means LOMAR is trained with the value of $\rho = x$. The average reward and competitive ratio are represented by AVG and CR, respectively — the higher, the better. The highest value in each testing setup is highlighted in bold. The AVG and CR for DRL are **12.909** and **0.544** respectively. The average reward for OPT is **13.209**.

We show the comparison of LOMAR with baseline algorithms in Table 1. First, we see that DRL has the highest average reward, but its empirical competitive ratio is the lowest. The expert algorithm Greedy is fairly robust, but has a lower average award than RL-based policies. Second, DRL-OS can improve the competitive ratio compared to DRL. But, its RL policy is trained alone without being aware of the online switching. Thus, by making the RL policy aware of online switching, LOMAR can improve the average reward compared to DRL-OS. Specifically, by training LOMAR using the same $\rho$ as testing it, we can obtain both the highest average cost and the highest competitive ratio. One exception is the minor decrease of competitive ratio when $\rho = 0.8$ for testing. This is likely due to the dataset and a few hard instances can affect the empirical competitive ratio, which also explains why the empirical competitive ratio is not necessarily monotonically increasing in the $\rho \in [0, 1]$. Nonetheless, unlike DRL that may only work well empirically without guarantees, LOMAR offers provable robustness guarantees while exploiting the power of RL to improve the average performance. The boxplots in Fig. 1 visualizes the reward ratio distribution of LOMAR, which further validates the importance of switching-aware training.

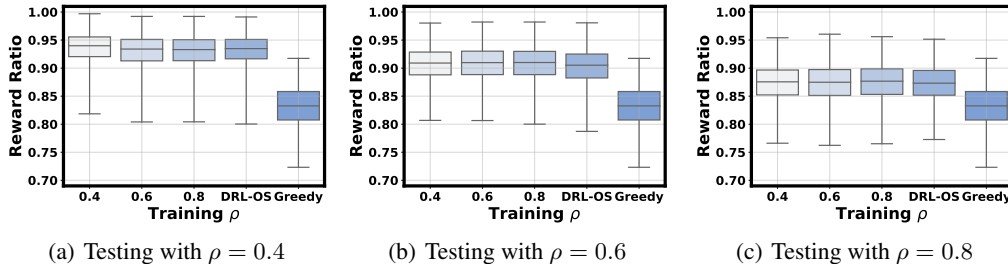

(a) Testing with $\rho = 0.4$     (b) Testing with $\rho = 0.6$     (c) Testing with $\rho = 0.8$

Figure 1: Boxplot for reward ratio with different $\rho$ within testing dataset. Greedy and DRL-OS are also shown here for comparison. The best average performance in each figure is achieved by choosing the same $\rho$ during training and testing.

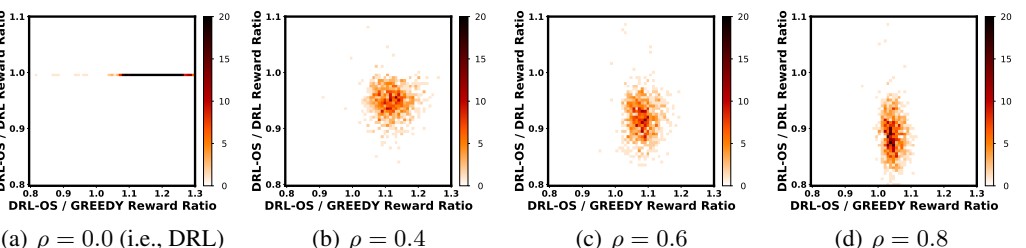

(a) $\rho = 0.0$ (i.e., DRL)    (b) $\rho = 0.4$    (c) $\rho = 0.6$    (d) $\rho = 0.8$

Figure 2: Histogram of bi-competitive reward ratios of DRL-OS (against Greedy and DRL) under different $\rho$.

To show the effect of switching with different $\rho$, we calculate the bi-competitive reward ratios. Specifically, for each problem instance, the bi-competitive ratio compares the actual reward against those of Greedy and RL model, respectively. To highlight the effect of online switching, we focus on DRL-OS (i.e., training the RL with $\rho = 0$) whose training process of RL model is not affected by $\rho$, because the RL model trained with $\rho > 0$ in LOMAR does not necessarily perform well on its own and the reward ratio of LOMAR to its RL model is not meaningful. The histogram of the bi-competitive ratios are visualized in Fig. 2. When $\rho = 0$, the ratio of DRL-OS/ DRL is always 1 unsurprisingly, since DRL-OS are essentially the same as DRL in this case (i.e., both trained and tested with $\rho = 0$). With a large $\rho$ (e.g. 0.8) for testing, the reward ratios of DRL-OS/Greedy for most samples are around 1, which means the robustness is achieved, as proven by our theoretical analysis. But on the other hand, DRL-OS has limited flexibility and can less exploit the good average performance of DRL. Thus, the hyperparameter $\rho \in [0, 1]$ governs the tradeoff between average performance and robustness relative to the expert and, like other hyperparameters, can be tuned to maximize the average performance subject to the robustness requirement.

We also consider a crowdsourcing application, as provided by the gMission dataset (Chen et al., 2014). Additional results for LOMAR and baselines in gMission are deferred to Appendix A.

## 7 CONCLUSION

In this paper, we propose LOMAR to achieve both good average-case and good worst-case performance for edge-weighted online bipartite matching. LOMAR includes a novel online switching operation to decide whether to follow the expert's decision or the RL decision for each online item arrival. We prove that for any $\rho \in [0, 1]$, LOMAR is $\rho$-competitive against any expert online algorithms, which directly translates a bounded competitive ratio against OPT if the expert algorithm itself has one. We also train the RL policy by explicitly considering the online switching operation so as to improve the average performance. Finally, we run empirical experiments to validate LOMAR.

## REPRODUCIBILITY STATEMENT

The details of proving Theorem 4.1 are included in Appendix C. The experimental codes and settings are based on the open-sourced resources in Alomrani et al. (2021). The implementation of `LOMAR` mainly includes adding the switching condition for training and testing based on the standard DRL-based algorithm Alomrani et al. (2021) and will be released upon publication.

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

APPENDIX

In the appendix, we show the experimental setup and additional results (Appendix A), algorithm details for the free-disposal setting (Appendix B), and finally the proof of Theorem 4.1 (Appendix C).

## A  EXPERIMENTAL SETTINGS AND ADDITIONAL RESULTS

Our implementation of all the considered algorithms, including LOMAR, is based on the source codes provided by Alomrani et al. (2021), which includes codes for training the RL model, data pre-proposing and performance evaluation. We conduct experiments on two real-world datasets: MovieLens Harper & Konstan (2015) and gMission Chen et al. (2014).

### A.1  MOVIELENS

#### A.1.1  SETUP AND TRAINING

We first sample $u_0$ movies from the original MovieLens dataset Harper & Konstan (2015). We then sample $v_0$ users and make sure each user can get at least one movie; otherwise, we remove the users that have no matched movies, and resample new users. After getting the topology graph, we use Gurobi to find the optimal matching decision. In our experiment, we set $u_0 = 10$ and $v_0 = 60$ to generate the training and testing datasets. The number of graph instances in the training and testing datasets are 20000 and 1000, respectively. For the sake of reproducibility and fair comparision, our settings follows the same setup of Alomrani et al. (2021). In particular, the general movie recommendation problem belongs to online submodular optimization, but it can actually be equivalently mapped to edge-weighted online bipartite matching with no free disposal under the setting considered in Alomrani et al. (2021). So by default, the capacity $c_u$ for each offline node is set as 1 and $w_{u,\max} = 5$. While LOMAR can use any RL architecture, we follow the design of *inv-ff-hist* proposed by Alomrani et al. (2021), which empirically demonstrates the best performance among all the considered architectures.

The input to our considered RL model is the edge weights $w_{uv}$ revealed by the online items plus some historical information, which includes: Mean and variances of each offline node's weights; Average degree of each offline nodes; Normalized step size; Percentage of offline nodes connected to the current node; Statistical information of these already matched nodes' weights (maximum, minimum, mean and variance); Ratio of matched offline node; Ratio of skips up to now; Normalized reward with respect to the offline node number. For more details of the historical information, readers are referred to Table 1 in Alomrani et al. (2021).

For applicable algorithms (i.e., DRL, DRL-OS, and LOMAR), we train the RL model for 300 epochs in the training dataset with a batch size of 100. In LOMAR, the parameter $B = 0$ is used to follow the strict definition of competitive ratio. We test the algorithms on the testing dataset to obtain the average reward and the worst-case competitive ratio empirically. By setting $\rho = 0$ for training, LOMAR is equivalent to the vanilla inv-ff-hist RL model (i.e., DRL) used in Alomrani et al. (2021). Using the same problem setup, we can reproduce the same results shown in Alomrani et al. (2021), which reaffirms the correctness of our data generation and training process.

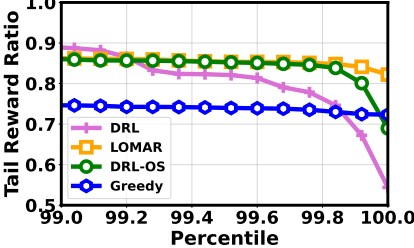

Figure 3: Tail reward ratio comparison. In this experiment, we set $\rho = 0.4$ for DRL-OS and LOMAR.

Additionally, training the RL model in LOMAR usually takes less than 8 hours on a shared research cluster with one NVIDIA K80 GPU, which is almost the same as the training the model for DRL in a standalone manner (i.e., setting $\rho = 0$ without considering online switching).

#### A.1.2  ADDITIONAL RESULTS

In Table 1, we have empirically demonstrated that LOMAR achieves the best tradeoff between the average reward and competitive ratio. In Fig 3, we further demonstrate that LOMAR not only achieves

a better worst-case competitive ratio (at 100.0%). The tail reward ratio of LOMAR is also good compared to the baseline algorithms. Specifically, we show the percentile of reward ratios (compared to the optimal offline algorithm) — the 100% means the worst-case empirical reward ratio (i.e., competitive ratio). We see that DRL has a bad high-percentile reward ratio and lacks performance robustness, although its lower-percentile cost ratio is better. This is consistent with the good average performance of LOMAR. Because of online switching, both DRL-OS and LOMAR achieve better robustness, and LOMAR is even better due to its awareness of the online switching operation during its training process. The expert Greedy has a fairly stable competitive ratio, showing its good robustness. But, it can be outperformed by other algorithms when we look at lower-percentile reward ratio.

### A.1.3 RESULTS FOR ANOTHER EXPERT ALGORITHM

Optimally competitive expert algorithms have been developed under the assumptions of random oder and/or i.i.d. rewards of different online items. In particular, by considering the random order setting, OSM (online secretary matching) has the optimal competitive ratio of $1/e$ (Kesselheim et al., 2013). Note that the competitive ratio for OSM is average over the random order of online items, while the rewards can be adversarially chosen. We show the empirical results in Fig. 4. As OSM skips the first $|\mathcal{V}|/e$ online items, it actually does not perform (in terms of the empirical worst-case cost ratio) as well as the default expert Greedy in our experiments despite its guaranteed competitive ratio against OPT. That said, we still observe the same trend as using Greedy for the expert: by tuning $\rho \in [0, 1]$, LOMAR achieves a good average performance while guaranteeing the competitiveness against the expert OSM (and against OPT as OSM itself is optimally competitive against OPT).

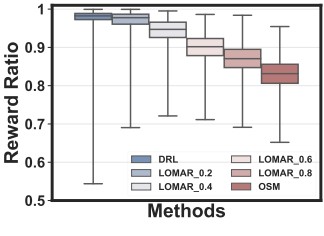
(a) Cost ratio against OPT

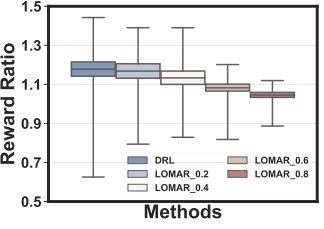
(b) Cost ratio against OSM

Figure 4: Cost ratio distribution (OSM as the expert)

Fig. 4 shows the empirical results in our testing dataset, which does not strictly satisfy the random order assumption required by OSM. Next, to satisfy the random order assumption, we select a typical problem instance and randomly vary the arrival orders of online items. We show the cost ratio averaged over the random arrival order in Table 2. Specifically, we calculate each cost ratio by 100 different random orders, and repeat this process 100 times. We show the mean and stand deviation of the average cost ratios (each averaged over 100 different random orders). We see that LOMAR improves the average cost ratio compared to OSM under the random order assumption. While DRL has a better average cost for this particular instance, it does not provide any guaranteed worst-case robustness as LOMAR.

|  | Pure ML | LOMAR $\rho = 0.2$ | LOMAR $\rho = 0.4$ | LOMAR $\rho = 0.6$ | LOMAR $\rho = 0.8$ | OSM |
|---|---|---|---|---|---|---|
| Mean | 0.9794 | 0.9688 | 0.9431 | 0.9095 | 0.8799 | 0.8459 |
| Std | 0.0074 | 0.0082 | 0.0078 | 0.0086 | 0.0084 | 0.0084 |

Table 2: Cost ratio (averaged over the random arrival order) for a typical graph instance

### A.2 GMISSION

The gMission dataset Chen et al. (2014) considers a crowdsourcing application, where the goal is to assign the tasks (online items) to workers (offline items). The edge weight between a certain online task and each worker can be calculated by the product of the task reward and the worker's success probability, which is determined by the physical location of workers and the type of tasks.

Our goal is to maximize the total reward given the capacity of each worker, which perfectly fits into our formulation in Eqn. equation 1.

We use the same data processing and RL architecture design as introduced in Section A.1.1. We train LOMAR with different $\rho$ in the gMission dataset by setting $u_0 = 10$, $v_0 = 60$, $w_{u,\max} = 1$. Again, we use Greedy as the expert, which is an empirically strong baseline algorithm as shown in Alomrani et al. (2021). Our results are all consistent with those presented in Alomrani et al. (2021).

### A.2.1 Testing on $10 \times 60$

In our the first result, we generate a testing dataset with $u_0 = 10$ and $v_0 = 60$, which is the same setting as our training dataset. In other words, the training and testing datasets have similar distributions. Specifically, Greedy's average reward and competitive ratio are 4.508 and 0.432, while these two values for DRL are 5.819 and 0.604, respectively. Thus, DRL performs outperforms Greedy in both average performance and the worst-case performance.

| | DRL-OS | | LOMAR $\rho = 0.4$ | | LOMAR $\rho = 0.6$ | | LOMAR $\rho = 0.8$ | | LOMAR $\rho = 0.9$ | |
|---|---|---|---|---|---|---|---|---|---|---|
| $\rho$ in Testing | AVG | CR | AVG | CR | AVG | CR | AVG | CR | AVG | CR |
| 0.4 | 5.553 | **0.599** | **5.573** | 0.598 | 5.553 | 0.598 | 5.523 | 0.598 | 5.535 | 0.598 |
| 0.6 | 5.389 | 0.591 | **5.429** | **0.619** | 5.420 | 0.619 | 5.403 | 0.623 | 5.402 | 0.623 |
| 0.8 | 5.102 | 0.543 | **5.115** | **0.543** | 5.111 | 0.523 | 5.110 | 0.521 | 5.107 | 0.521 |
| 0.9 | 4.836 | 0.495 | 4.836 | 0.495 | 4.839 | 0.495 | 4.839 | 0.540 | **4.839** | **0.540** |

Table 3: Performance comparison in gMission $10 \times 60$ for different $\rho$. LOMAR with $\rho = y$ means LOMAR is trained with $\rho = y$.

Next, we show the results for LOMAR and DRL-OS under different $\rho \in [0, 1]$ in Table 3. In general, by setting a larger $\rho$ for inference, both LOMAR and DRL-OS are closer to the expert algorithm Greedy, because there is less freedom for the RL decisions. As a result, when $\rho$ increases for inference, the average rewards of both DRL-OS and LOMAR decrease, although they have guaranteed robustness whereas DRL does not. Moreover, by training the RL model with explicit awareness of online switching, LOMAR can have a higher average cost than DRL-OS, which reconfirms the benefits of training the RL model by considering its downstream operation. Interestingly, by setting $\rho$ identical for both training and testing, the average reward may not always be the highest for LOMAR. This is partially because of the empirical testing dataset. Another reason is that, in this test, DRL alone already performs the best (both on average and in the worst case). Hence, by setting a smaller $\rho$ for inference, LOMAR works better empirically though it is trained under a different $\rho$. Nonetheless, this does not void the benefits of guaranteed robustness in LOMAR. The empirically better performance of DRL lacks guarantees, which we show as follows.

### A.2.2 Testing on $100 \times 100$

In our second test, we consider an opposite case compared to the first one. We generate a testing dataset with $u_0 = 100$ and $v_0 = 100$, which is different from the training dataset setting. As a result, the training and testing datasets have very different distributions, making DRL perform very badly. Specifically, Greedy's average reward and competitive ratio are 40.830 and 0.824, and these two values for DRL are 32.938 and 0.576, respectively. DRL has an even lower average reward than Greedy, showing its lack of performance robustness.

We show the results for LOMAR and DRL-OS under different $\rho \in [0, 1]$ in Table 4. In general, by setting a larger $\rho$ for inference, both LOMAR and DRL-OS are closer to the expert algorithm Greedy. As Greedy works empirically much better than DRL in terms of the average performance and the worst-case performance, both LOMAR and DRL-OS have better performances when we increase $\rho$ to let Greedy safeguard the RL decisions more aggressively. Moreover, by training the RL model with explicit awareness of online switching, LOMAR can have a higher average cost than DRL-OS, which further demonstrates the benefits of training the RL model by considering its downstream operation. Also, interestingly, by setting $\rho$ identical for both training and testing, the average reward may not be the highest for LOMAR, partially because of the empirical testing dataset. Another reason is that, in this test, DRL alone already performs very badly (both on average and in the worst case)

|  | DRL-OS | | LOMAR $\rho = 0.4$ | | LOMAR $\rho = 0.6$ | | LOMAR $\rho = 0.8$ | | LOMAR $\rho = 0.9$ | |
|---|---|---|---|---|---|---|---|---|---|---|
| $\rho$ in Testing | AVG | CR | AVG | CR | AVG | CR | AVG | CR | AVG | CR |
| 0.4 | 33.580 | 0.604 | 37.030 | 0.707 | 38.199 | 0.750 | 38.324 | 0.750 | **38.538** | **0.766** |
| 0.6 | 34.973 | 0.680 | 37.490 | 0.731 | 38.518 | 0.762 | 38.505 | 0.756 | **38.727** | **0.767** |
| 0.8 | 37.939 | 0.758 | 38.866 | 0.775 | 39.502 | 0.782 | 39.385 | **0.794** | **39.552** | 0.781 |
| 0.9 | 39.772 | 0.794 | 40.057 | 0.803 | **40.377** | 0.806 | 40.239 | **0.812** | 40.332 | 0.798 |

Table 4: Performance comparison on gMission $100 \times 100$ for different $\rho$. LOMAR with $\rho = y$ means LOMAR is trained with $\rho = y$.

due to the significant training-testing distributional discrepancy. Hence, by setting a higher $\rho$, LOMAR works better empirically though it is tested under a different $\rho$. An exception is when testing LOMAR with $\rho = 0.9$: setting $\rho = 0.6/0.8$ for training makes LOMAR perform slightly better in terms of the average performance and worst-case performance, respectively. But, setting $\rho = 0.9$ for training still brings benefits to LOMAR compared to DRL-OS that does not consider the downstream online switching operation.

*To sum up*, our experimental results under different settings demonstrate: LOMAR's empirical improvement in terms of the average reward compared to DRL-OS; the improved competitive ratio of LOMAR and DRL-OS compared to DRL, especially when the training-testing distributions differ significantly; and the improved average reward of LOMAR compared to Greedy when RL is good. Therefore, LOMAR can exploit the power of RL while provably guaranteeing the performance robustness.

## B  FREE DISPOSAL

The offline version of bipartite matching with free disposal can be expressed as:

$$\textbf{With Free Disposal:} \quad \max_{x_{uv} \in \{0,1\}, u \in \mathcal{U}} \sum \left( \max_{\mathcal{S} \subseteq \mathcal{V}_u, |\mathcal{S}| \leq c_u} \sum_{v \in \mathcal{S}} w_{uv} \right) \tag{5}$$
$$\text{s.t.} \quad \mathcal{V}_u = \{v \in \mathcal{V} \mid x_{uv} = 1\} \ \forall u \in \mathcal{U}, \quad \sum_{u \in \mathcal{U}} x_{uv} \leq 1, \ \forall v \in \mathcal{V},$$

where $\mathcal{V}_u = \{v \in \mathcal{V} \mid x_{uv} = 1\}$ is the set of online items matched to $u \in \mathcal{U}$ and the objective $\max_{\mathcal{S} \in \mathcal{V}_u, |\mathcal{S}| \leq c_u} \sum_{v \in \mathcal{S}} x_{uv} w_{uv}$ indicates that only up to top $c_u$ rewards are counted for $u \in \mathcal{U}$.

In the free-disposal setting, it is more challenging to design the switching conditions to guarantee the robustness. The reason is the additional flexibility allowed for matching decisions — each offline item $u \in \mathcal{U}$ is allowed to be matched more than $c_u$ times although only up to top $c_u$ rewards actually count Mehta (2013); Fahrbach et al. (2020). For example, even though LOMAR and the expert assign the same number of online items to an offline item $u \in \mathcal{U}$ and LOMAR is better than the expert at a certain step, future high-reward online items can still be assigned to $u \in \mathcal{U}$, increasing the expert's total reward or even equalizing the rewards of LOMAR and the expert (i.e., high-reward future online items become the top $c_u$ items for $u \in \mathcal{U}$ for both LOMAR and the expert). Thus, the temporarily "higher" rewards received by LOMAR must be hedged against such future uncertainties. Before designing our switching condition for the free-disposal setting, we first define the set containing the top $c_u$ online items for $u \in \mathcal{U}$ after assignment of $v$:

$$\mathcal{E}_{u,v}(\mathcal{V}_{u,v}) = \arg \max_{\mathcal{E} \subseteq \mathcal{V}_{u,v}, |\mathcal{E}| = c_u} \sum_{v \in \mathcal{E}} w_{uv}, \tag{6}$$

where $\mathcal{V}_{u,v}$ is the set of all online items matched to $u \in \mathcal{U}$ so far after assignment of item $v \in \mathcal{V}$. When there are fewer than $c_u$ items in $\mathcal{V}_{uv}$, we will simply add null items with reward 0 to $\mathcal{E}_{u,v}$ such that $|\mathcal{E}_{u,v}| = c_u$. We also sort the online items denoted as $e_{u,i}$, for $i = 1, \cdots, c_u$, contained in $\mathcal{E}_{u,v}$ according to their weights in an increasing order such that $w_{u,e_{u,1}} \leq \cdots \leq w_{u,e_{u,c_u}}$. Similarly, we define the same top-$c_u$ item set for the expert algorithm $\pi$ by adding the superscript $\pi$.

Next, we define the following value which indicates the maximum possible additional reward for the expert algorithm $\pi$ if LOMAR simply switches to the expert and follows it for all the future steps

---

**Algorithm 2** Inference of `LOMAR` (Free Disposal)

---

1: **Initialization:** The actual set of items matched to $u \in \mathcal{U}$ is $\mathcal{V}_{u,v}$ after sequentially-arriving item $v$'s assignment with $\mathcal{V}_{u,0} = \emptyset$, the actual remaining capacity is $b_u = c_u$ for $u \in \mathcal{U}$, and the actual cumulative reward is $R_0 = \sum_{u \in \mathcal{U}} f_u(\mathcal{V}_{u,0}) = 0$. The same notations apply to the expert algorithm $\pi$ by adding the superscript $\pi$. Competitive ratio requirement $\rho \in [0, 1]$ and slackness $B \geq 0$ with respect to the expert algorithm $\pi$.
2: **for** $v = 1$ to $|\mathcal{V}|$ **do**
3:     Run the algorithm $\pi$ and match the item $v$ to $u \in \mathcal{U}$ based on the expert's decision $u = x_v^\pi$.
4:     Update the expert's decision set and reward for offline item $u = x_v^\pi$:         $\mathcal{V}_{x_v^\pi,v}^\pi = \mathcal{V}_{x_v^\pi,v-1}^\pi \bigcup \{v\}$ and $f_{x_v^\pi} = f_{x_v^\pi}(\mathcal{V}_{x_v^\pi,v}^\pi)$.
5:     Update the expert's cumulative reward $R_v^\pi = \sum_{u \in \mathcal{U}} f_u$
6:     **for** $u$ in $\mathcal{U}$ **do**
7:         Collect the available history information $I_u$ about item $u$
8:         Run the RL model to get score: $s_u = w_{uv} - h_\theta(I_u, w_{uv})$ where $\theta$ is the network weight
9:     **end for**
10:    Calculate the probability of choosing each item $u$: $\{\{\tilde{s}_u\}_{u \in \mathcal{U}}\} = \text{softmax}\{\{s_u\}_{u \in \mathcal{U}}\}$.
11:    Obtain RL decision: $\tilde{x}_v = \arg\max_{u \in \mathcal{U} \bigcup \{\text{skip}\}} \{\{\tilde{s}_u\}_{u \in \mathcal{U}}\}$.
12:    Find $\Delta f_{\tilde{x}_v}$ in Eqn. equation 8 and $G\left(\tilde{x}_v, \{\mathcal{V}_{u,v-1}\}_{u \in \mathcal{U}}, \{\mathcal{V}_{u,v}^\pi\} u \in \mathcal{U}\right)$ in Eqn. equation 14
13:    **if** $R_{v-1} + \Delta f_{\tilde{x}_v} \geq \rho\left(R_v^\pi + G\left(\tilde{x}_v, \{\mathcal{V}_{u,v-1}\}_{u \in \mathcal{U}}, \{\mathcal{V}_{u,v}^\pi\}_{u \in \mathcal{U}}\right)\right) - B$ **then**
14:       Select $x_v = \tilde{x}_v$.   //Follow the ML action
15:    **else**
16:       Select $x_v = x_v^\pi$.   //Follow the expert
17:    **end if**
18:    Update assignment and reward: $\mathcal{V}_{x_v,v} = \mathcal{V}_{x_v,v-1} \bigcup \{v\}$ and $R_v = \sum_{u \in \mathcal{U}} f_u(\mathcal{V}_{u,v})$
19: **end for**

---

$v + 1, v + 2, \cdots$:

$$G\left(\tilde{x}_v, \{\mathcal{V}_{u,v-1}\}_{u \in \mathcal{U}}, \{\mathcal{V}_{u,v}^\pi\}_{u \in \mathcal{U}}\right) = \sum_{u \in \mathcal{U}} \left(\max_{i=1,\cdots,c_u} \sum_{j=1}^{i}(w_{u,e_{u,j}} - w_{u,e_{u,j}^\pi})\right)^+, \qquad (7)$$

where $e_{u,j}^\pi \in \mathcal{E}_u^\pi(\mathcal{V}_{u,v}^\pi)$, and $e_{u,j} \in \mathcal{E}_u(\tilde{\mathcal{V}}_{u,v})$ in which $\tilde{\mathcal{V}}_{u,v} = \mathcal{V}_{u,v-1}$ if $\tilde{x}_v \neq u$ and $\tilde{\mathcal{V}}_{u,v} = \mathcal{V}_{u,v-1} \bigcup \{v\}$ if $\tilde{x}_v = u$.

The interpretation is as follows. Suppose that `LOMAR` follows the RL decision for online item $v$. If it has a higher cumulative reward for the $j$-th item in the top-$c_u$ item set $\mathcal{E}_{u,v}$ than the expert algorithm $\pi$, then the expert can still possibly offset the reward difference $w_{u,e_{u,j}} - w_{u,e_{u,j}^\pi}$ by receiving a high-reward future online item that replaces the $j$-th item for both `LOMAR` and the expert. Nonetheless, in the free-disposal model, the items in the top-$c_u$ set $\mathcal{E}_{u,v}$ are removed sequentially — the lowest-reward item will be first removed from the sorted set $\mathcal{E}_{u,v}$, followed by the next lowest-reward item, and so on. Thus, in order for a high-reward item to replace the $i$-th item in the sorted set $\mathcal{E}_{u,v}$, the first $(i-1)$ items have to be removed first by other high-reward online items. As a result, if `LOMAR` has a lower reward for the $j$-th item (for $j \leq i$) in the top-$c_u$ item set $\mathcal{E}_{u,v}$ than the expert algorithm $\pi$, then it will negatively impact the expert's additional reward gain in the future. Therefore, for item $u \in \mathcal{U}$ we only need to find the highest total reward difference, $\left(\max_{i=1,\cdots,c_u} \sum_{j=1}^{i}(w_{u,e_{u,j}} - w_{u,e_{u,j}^\pi})\right)^+$, that can be offset for the expert algorithm $\pi$ by considering that $i$ items are replaced by future high-reward online items for $i = 1, \cdots, c_u$. If $\max_{i=1,\cdots,c_u} \sum_{j=1}^{i}(w_{u,e_{u,j}} - w_{u,e_{u,j}^\pi})$ is negative (i.e., the expert algorithm cannot possibly gain higher rewards than `LOMAR` by receiving high-reward online items to replace its existing ones), then we use $0$ as the hedging reward.

Finally, by summing up the hedging rewards for all the offline items $u \in \mathcal{U}$, we obtain the total hedging reward in Eqn. equation 7. Based on this hedging reward, we have the condition (Line 28 in Algorithm 1 for `LOMAR` to follow the RL decision: $R_{v-1} + \Delta f_{\tilde{x}_v} \geq \rho\left(R_v^\pi + G\left(\tilde{x}_v, \{\mathcal{V}_{u,v-1}\}_{u \in \mathcal{U}}, \{\mathcal{V}_{u,v}^\pi\}_{u \in \mathcal{U}}\right)\right) - B$, where $\Delta f_{\tilde{x}_v}$ defined below is the additional reward that would be obtained by following the RL decision:

$$\Delta f_{\tilde{x}_v} = f_{\tilde{x}_v}(\mathcal{V}_{\tilde{x}_v,v} \bigcup \{v\}) - f_{\tilde{x}_v}(\mathcal{V}_{\tilde{x}_v,v-1}), \qquad (8)$$

---

**Algorithm 3** Training `LOMAR` with Online Switching (No Free Disposal)

---

**Input:** Available history information $\{I_u \mid u \in \mathcal{U}\}$; set of weights for the online item arrival $\{w_{uv}, u \in \mathcal{U}\}$; up to the online item $v$, the expert's cumulative reward $R_v^\pi$, expert's set of items matched to $u \in \mathcal{U}$ is $\mathcal{V}_{u,v}^\pi$; up to the online node $v-1$, the actual cumulative reward $R_{v-1}$, the actual set of items matched to $u \in \mathcal{U}$ is $\mathcal{V}_{u,v-1}$;

1: **for** $u$ in $\mathcal{U}_a$ **do**
2:    $s_u = w_{uv} - h_\theta(I_u, w_{uv})$
3: **end for**
4: Calculate the probabilities of selecting an available item or skip:
   $\{\{\tilde{s}_u\}_{u \in \mathcal{U}_a}, \tilde{s}_{\text{skip}}\} = \text{softmax}\{\{s_u\}_{u \in \mathcal{U}_a}, 0\}$.
5: Obtain the RL action: $\tilde{x}_v = \arg\max_{\mathcal{U}_a \bigcup\{\text{skip}\}}\{\{\tilde{s}_u\}_{u \in \mathcal{U}_a}, \tilde{s}_{\text{skip}}\}$.
6: Calculate $R_{diff}$ based on the switching condition
$$R_{diff} = R_{v-1} + w_{\tilde{x}_v, v} - \rho\left(R_v^\pi + \sum_{u \in \mathcal{U}}\left(|\mathcal{V}_{u,v-1}| - |V_{u,v}^\pi| + \mathbb{I}_{u=\tilde{x}_v}\right)^+ \cdot w_{u,\max}\right) + B$$
7: **if** $x_v^\pi \in \mathcal{U}_a$ **then**
8:    The probability of each action is $\left\{\{\tilde{s}_u^\pi\}_{u \in \mathcal{U}_a}, \tilde{s}_{\text{skip}}^\pi\right\} = \{\{1\}_{u=x_v^\pi}, \{0\}_{u \neq x_v^\pi, \text{skip}}\}$
9: **else**
10:    The probability of each action is $\left\{\{\tilde{s}_u^\pi\}_{u \in \mathcal{U}_a}, \tilde{s}_{\text{skip}}^\pi\right\} = \{\{0\}_{u \in \mathcal{U}_a}, 1\}$
11: **end if**
12: With online switching, the probabilities of following RL or expert are
   $\{\tilde{s}_{OS}, \tilde{s}_{OS}^\pi\} = \text{softmax}\{R_{diff}/t, 0\}$
13: Calibrated probabilities of choosing an available offline item $u$ or skip are
$$\{\{\hat{s}_u\}_{u \in \mathcal{U}_a}, \hat{s}_{\text{skip}}\} = \tilde{s}_{OS} \cdot \{\{\tilde{s}_u\}_{u \in \mathcal{U}_a}, \tilde{s}_{\text{skip}}\} + \tilde{s}_{OS}^\pi \cdot \left\{\{\tilde{s}_u^\pi\}_{u \in \mathcal{U}_a}, \tilde{s}_{\text{skip}}^\pi\right\}.$$

---

in which $f_u = f_u(\mathcal{V}') = \max_{\mathcal{S} \in \mathcal{V}', |\mathcal{S}| \leq c_u} \sum_{v \in \mathcal{S}} w_{uv}$ is the reward function for an offline item $u \in \mathcal{U}$ in the free-disposal model. The condition means that if `LOMAR` can maintain the competitive ratio $\rho$ against the expert algorithm $\pi$ by being able to hedge against any future uncertainties even in the worst case, then it can safely follow the RL decision $\tilde{x}_v$ at step $v$.

**Training with free disposal.** The training process for the free-disposal setting is the same as that for the no-free-disposal setting, except for Line 6 of Algorithm 3 in which we need to modify $R_{diff}$ based on the switching condition (i.e., Line 13 of Algorithm 2) for the free-disposal setting.

## C   PROOF OF THEOREM 4.1

The key idea of proving Theorem 4.1 is to show that there always exist feasible actions (either following the expert or skip) while being able to guarantee the robustness if we follow the switching condition. Next, we prove Theorem 4.1 for the no-free-disposal and free-disposal settings, respectively.

### C.1   NO FREE DISPOSAL

Denote $\mathcal{V}_{u,v}$ as the actual set of items matched to $u \in \mathcal{U}$ after making decision for $v$. Denote $\mathcal{V}_{u,v}^\pi$ as the expert's set of items matched to $u \in \mathcal{U}$. We first prove a technical lemma.

**Lemma C.1.** *Assuming that the robustness condition is met after making the decision for $v - 1$, i.e.* $R_{v-1} \geq \rho\left(R_{v-1}^\pi + \sum_{u \in \mathcal{U}}\left(|\mathcal{V}_{u,v-1}| - |V_{u,v-1}^\pi|\right)^+ \cdot w_{u,\max}\right) - B$. *If at the step when $v$ arrives and the expert's decision $x_v^\pi$ is not available for matching, then $x_v = $ skip always satisfies* $R_v \geq \rho\left(R_v^\pi + \sum_{u \in \mathcal{U}}\left(|\mathcal{V}_{u,v}| - |V_{u,v}^\pi|\right)^+ \cdot w_{u,\max}\right) - B$.

*Proof.* If the item $x_v^\pi$ is not available for matching, it must have been consumed before $v$ arrives, which means $|\mathcal{V}_{x_v^\pi, v-1}| - |V_{x_v^\pi, v-1}^\pi| \geq 1$ (since otherwise the expert cannot choose $x_v^{pi}$ either). Since $x_v = $ skip, we have $R_v = R_{v-1}$ and $\mathcal{V}_{u,v} = \mathcal{V}_{u,v-1}, \quad \forall u \in \mathcal{U}$. Then, by the robustness assumption

of the previous step, we have

$$R_v = R_{v-1} \geq \rho \left( R_{v-1}^\pi + \sum_{u \in \mathcal{U}} \left( |\mathcal{V}_{u,v-1}| - |V_{u,v-1}^\pi| \right)^+ \cdot w_{u,\max} \right) - B$$

$$\geq \rho \left( R_{v-1}^\pi + w_{x_v^\pi, v} - w_{x_v^\pi, \max} + \sum_{u \in \mathcal{U}} \left( |\mathcal{V}_{u,v-1}| - |V_{u,v-1}^\pi| \right)^+ \cdot w_{u,\max} \right) - B \quad (9)$$

$$= \rho \left( R_v^\pi + \sum_{u \in \mathcal{U}} \left( |\mathcal{V}_{u,v}| - |V_{u,v}^\pi| \right)^+ \cdot w_{u,\max} \right) - B$$

where the last equality holds because $(|\mathcal{V}_{u,v}| - |\mathcal{V}_{u,v}^\pi|)^+ - (|\mathcal{V}_{u,v-1}| - |\mathcal{V}_{u,v-1}^\pi|)^+ = -1$ if $u = x_v^\pi$, and $(|\mathcal{V}_{u,v}| - |\mathcal{V}_{u,v}^\pi|)^+ - (|\mathcal{V}_{u,v-1}| - |\mathcal{V}_{u,v-1}^\pi|)^+ = 0$ otherwise. □

We next prove by induction that the condition

$$R_v \geq \rho \left( R_v^\pi + \sum_{u \in \mathcal{U}} \left( |\mathcal{V}_{u,v}| - |V_{u,v}^\pi| \right)^+ \cdot w_{u,\max} \right) - B \quad (10)$$

holds for all steps by Algorithm 1.

At the first step, if $\tilde{x}_v$ is not the same as $x_v^\pi$ and $w_{\tilde{x}_v, v} \geq \rho \left( w_{x_v^\pi, v} + w_{u,\max} \right) - B$, we select the RL decision $x_v = \tilde{x}_v$, and the robustness condition equation 10 is satisfied. Otherwise, we select the expert action $x_v = x_v^\pi$ and the condition still holds since $A_v = C_v$ and $w_{x_v, v} \geq \rho w_{x_v^\pi, v} - B$ holds when $\rho \leq 1$ and $B \geq 0$.

Then, assuming that the robustness condition in equation 10 is satisfied after making the decision for $v - 1$, we need to prove it is also satisfied after making the decision for $v$. If the condition in equation 3 in Algorithm 1 is satisfied, then $x_v = \tilde{x}_v$ and so equation 10 holds naturally. Otherwise, if the expert action $x_v^\pi$ is available for matching, then we select expert action $x_v = x_v^\pi$. Then, we have $w_{x_v, v} \geq \rho w_{x_v^\pi, v} - B$ and $|\mathcal{V}_{u,v}| - |V_{u,v}^\pi| = |\mathcal{V}_{u,v-1}| - |V_{u,v-1}^\pi|, \quad \forall u \in \mathcal{U}$, and hence the condition equation 10 still holds. Other than these two cases, we also have the option to "skip", i.e. $x_v = \text{skip}$. By Lemma C.1, the condition equation 10 still holds. Therefore, we prove that the condition equation 10 holds for every step.

After the last step $v = |\mathcal{V}|$, we must have

$$R_v \geq \rho \left( R_v^\pi + \sum_{u \in \mathcal{U}} \left( |\mathcal{V}_{u,\hat{v}}| - |V_{u,\hat{v}}^\pi| \right)^+ \cdot w_{u,\max} \right) - B \geq \rho R_v^\pi - B \quad (11)$$

where $R_v$ and $R_v^\pi$ are the total rewards of LOMAR and the expert algorithm $\pi$ after the last step $v = |\mathcal{V}|$, respectively. This completes the proof for the no-free-disposal case.

## C.2 WITH FREE DISPOSAL

We now turn to the free-disposal setting which is more challenging than the no-free-disposal setting because of the possibility of using future high-reward items to replace existing low-reward ones.

We first denote $\Delta f_{x_v^\pi}$ as the actual additional reward obtained by following the expert's decision $x_v^\pi$,

$$\Delta f_{x_v^\pi} = f_{x_v^\pi}(\mathcal{V}_{x_v^\pi, v} \bigcup \{v\}) - f_{x_v^\pi}(\mathcal{V}_{x_v^\pi, v-1}), \quad (12)$$

Additionally, we denote $\Delta f_{x_v^\pi}^\pi$ as the expert's additional reward of choosing $x_v^\pi$, where

$$\Delta f_{x_v^\pi}^\pi = f_{x_v^\pi}(\mathcal{V}_{x_v^\pi, v}^\pi \bigcup \{v\}) - f_{x_v^\pi}(\mathcal{V}_{x_v^\pi, v-1}^\pi). \quad (13)$$

For presentation convenience, we rewrite the hedging reward as $\tilde{G}\left(\{\mathcal{V}_{u,v}\}_{u \in \mathcal{U}}, \{\mathcal{V}_{u,v}^\pi\}_{u \in \mathcal{U}}\right)$ as

$$\tilde{G}\left(\{\mathcal{V}_{u,v}\}_{u \in \mathcal{U}}, \{\mathcal{V}_{u,v}^\pi\}_{u \in \mathcal{U}}\right) = \sum_{u \in \mathcal{U}} \left( \max_{i=1,\cdots,c_u} \sum_{j=1}^{i} (w_{u,e_{u,j}} - w_{u,e_{u,j}^\pi}) \right)^+, \quad (14)$$

where $e_{u,j}^\pi \in \mathcal{E}_u^\pi(\mathcal{V}_{u,v}^\pi)$, $e_{u,j} \in \mathcal{E}_u(\mathcal{V}_{u,v})$, and $\mathcal{E}_u$ is defined in Eqn. equation 6.

**Lemma C.2.** *Assuming that the robustness condition is met after making the decision for $v-1$, i.e.*
$R_{v-1} \geq \rho\left(R_{v-1}^\pi + \tilde{G}\left(\{\mathcal{V}_{u,v-1}\}_{u\in\mathcal{U}}, \{\mathcal{V}_{u,v}^\pi\}_{u\in\mathcal{U}}\right)\right) - B$. *At step $v$, we have $\Delta f_{x_v^\pi} - \Delta f_{x_v^\pi}^\pi \geq$*
$G\left(x_v^\pi, \{\mathcal{V}_{u,v-1}\}_{u\in\mathcal{U}}, \{\mathcal{V}_{u,v}^\pi\}_{u\in\mathcal{U}}\right) - \tilde{G}\left(\{\mathcal{V}_{u,v-1}\}_{u\in\mathcal{U}}, \{\mathcal{V}_{u,v-1}^\pi\}_{u\in\mathcal{U}}\right)$.

*Proof.* We begin with "$G\left(x_v^\pi, \{\mathcal{V}_{u,v-1}\}_{u\in\mathcal{U}}, \{\mathcal{V}_{u,v}^\pi\}_{u\in\mathcal{U}}\right) - \tilde{G}\left(\{\mathcal{V}_{u,v-1}\}_{u\in\mathcal{U}}, \{\mathcal{V}_{u,v-1}^\pi\}_{u\in\mathcal{U}}\right)$" in
Lemma C.2. By definition, it can be written as

$$
\begin{aligned}
&G\left(x_v^\pi, \{\mathcal{V}_{u,v-1}\}_{u\in\mathcal{U}}, \{\mathcal{V}_{u,v}^\pi\}_{u\in\mathcal{U}}\right) - \tilde{G}\left(\{\mathcal{V}_{u,v-1}\}_{u\in\mathcal{U}}, \{\mathcal{V}_{u,v-1}^\pi\}_{u\in\mathcal{U}}\right) \\
&= \left(\max_{i=1,\cdots,c_u}\sum_{j=1}^{i}(w_{u,\hat{e}_{u,j}} - w_{u,\hat{e}_{u,j}^\pi})\right)^+ - \left(\max_{i=1,\cdots,c_u}\sum_{j=1}^{i}(w_{u,e_{u,j}} - w_{u,e_{u,j}^\pi})\right)^+
\end{aligned}
\tag{15}
$$

where $u = x_v^\pi$, $\hat{e}_{u,j}^\pi \in \mathcal{E}_u^\pi(\mathcal{V}_{u,v-1}^\pi \bigcup\{v\})$, and $\hat{e}_{u,j} \in \mathcal{E}_u(\mathcal{V}_{u,v-1}\bigcup\{v\})$. Besides, $e_{u,j}^\pi \in \mathcal{E}_u^\pi(\mathcal{V}_{u,v-1}^\pi)$, and $e_{u,j} \in \mathcal{E}_u(\mathcal{V}_{u,v-1})$.

To prove the lemma, we consider four possible cases for $w_{u,v}$ to cover all the cases.

**Case 1**: If the reward for $v$ is small enough such that $w_{u,v} < w_{u,e_{u,1}}$ and $w_{u,v} < w_{u,e_{u,1}^\pi}$, then
$v \notin \mathcal{E}_u(\mathcal{V}_{u,v-1}\bigcup\{v\})$ and $v \notin \mathcal{E}_u(\mathcal{V}_{u,v-1}^\pi \bigcup\{v\})$. Then we have $\Delta f_{x_v^\pi} = \Delta f_{x_v^\pi}^\pi = 0$, since both the
expert and LOMAR cannot gain any reward from the online item $v$. From Eqn. equation 15, we can
find that the right-hand side is also 0. Therefore, the conclusion in Lemma C.2 holds with the equality
activated.

**Case 2**: If the reward for $v$ is large enough such that $w_{u,v} > w_{u,e_{u,1}}$ and $w_{u,v} > w_{u,e_{u,1}^\pi}$, then
$v \in \mathcal{E}_u(\mathcal{V}_{u,v-1}\bigcup\{v\})$ and $v \in \mathcal{E}_u(\mathcal{V}_{u,v-1}^\pi\bigcup\{v\})$. In other words, we will remove the smallest-
reward item $e_{u,1} \notin \mathcal{E}_u(\mathcal{V}_{u,v-1}\bigcup\{v\})$ and $e_{u,1}^\pi \notin \mathcal{E}_u(\mathcal{V}_{u,v-1}^\pi\bigcup\{v\})$. Then

$$
G\left(x_v^\pi, \{\mathcal{V}_{u,v-1}\}_{u\in\mathcal{U}}, \{\mathcal{V}_{u,v}^\pi\}_{u\in\mathcal{U}}\right) - \tilde{G}\left(\{\mathcal{V}_{u,v-1}\}_{u\in\mathcal{U}}, \{\mathcal{V}_{u,v-1}^\pi\}_{u\in\mathcal{U}}\right) \leq -w_{u,e_{u,1}} + w_{u,e_{u,1}^\pi}
$$

The inequality holds because $(w_{u,e_{u,1}} - w_{u,e_{u,1}^\pi})^+ \geq w_{u,e_{u,1}} - w_{u,e_{u,1}^\pi}$. In this case, $\Delta f_{x_v^\pi} = w_{u,v} - w_{u,e_{u,1}}$ and $\Delta f_{x_v^\pi}^\pi = w_{u,v} - w_{u,e_{u,1}^\pi}$. Therefore, the conclusion in Lemma C.2 holds.

**Case 3**: If the reward for $v$ satisfies $w_{u,v} \geq w_{u,e_{u,1}}$ and $w_{u,v} \leq w_{u,e_{u,1}^\pi}$, then $v \in \mathcal{E}_u(\mathcal{V}_{u,v-1}\bigcup\{v\})$
and $v \notin \mathcal{E}_u(\mathcal{V}_{u,v-1}^\pi\bigcup\{v\})$. In other words, even if $v \in \mathcal{E}_u(\mathcal{V}_{u,v-1}\bigcup\{v\})$ (i.e., the online item $v$
produces additional rewards for LOMAR), the reward of $v$ is still smaller than the smallest reward for
the expert. Then, we have $G\left(x_v^\pi, \{\mathcal{V}_{u,v-1}\}_{u\in\mathcal{U}}, \{\mathcal{V}_{u,v}^\pi\}_{u\in\mathcal{U}}\right) - \tilde{G}\left(\{\mathcal{V}_{u,v-1}\}_{u\in\mathcal{U}}, \{\mathcal{V}_{u,v-1}^\pi\}_{u\in\mathcal{U}}\right) = 0$. In this case, $\Delta f_{x_v^\pi} = w_{u,v} - w_{u,e_{u,1}} \geq 0$ and $\Delta f_{x_v^\pi}^\pi = 0$. Therefore, the conclusion in Lemma C.2
still holds.

**Case 4**: If the reward for $v$ satisfies $w_{u,v} \leq w_{u,e_{u,1}}$ and $w_{u,v} \geq w_{u,e_{u,1}^\pi}$, then in this case, only the
current smallest-reward item is replaced with $v$ for the expert, while the reward of LOMAR remains
unchanged. Thus, we have

$$
G\left(x_v^\pi, \{\mathcal{V}_{u,v-1}\}_{u\in\mathcal{U}}, \{\mathcal{V}_{u,v}^\pi\}_{u\in\mathcal{U}}\right) - \tilde{G}\left(\{\mathcal{V}_{u,v-1}\}_{u\in\mathcal{U}}, \{\mathcal{V}_{u,v-1}^\pi\}_{u\in\mathcal{U}}\right) = w_{u,e_{u,1}^\pi} - w_{u,v}.
$$

In this case, $\Delta f_{x_v^\pi} = 0$ and $\Delta f_{x_v^\pi}^\pi = w_{u,v} - w_{u,e_{u,1}^\pi}$. Then the conclusion in Lemma C.2 still holds
with the equality activated. $\qquad\square$

We next prove by induction that the condition

$$
R_v \geq \rho\left(R_v^\pi + \tilde{G}\left(\{\mathcal{V}_{u,v}\}_{u\in\mathcal{U}}, \{\mathcal{V}_{u,v}^\pi\}_{u\in\mathcal{U}}\right)\right) - B
\tag{16}
$$

holds for all steps by Algorithm 1.

At the first step, by using $x_v = x_v^\pi$, we have $R_v = R_v^\pi$ and $\tilde{G}\left(\{\mathcal{V}_{u,v}\}_{u\in\mathcal{U}}, \{\mathcal{V}_{u,v}^\pi\}_{u\in\mathcal{U}}\right) = 0$, and it
is obvious that the condition in equation 16 is satisfied. Thus, there is at least one solution $x_v = x_v^\pi$
for our robustness condition in equation 16.

Starting from the second step, assume that after the step $v-1$, we already have

$$
R_{v-1} \geq \rho\left(R_{v-1}^\pi + \tilde{G}\left(\{\mathcal{V}_{u,v-1}\}_{u\in\mathcal{U}}, \{\mathcal{V}_{u,v-1}^\pi\}_{u\in\mathcal{U}}\right)\right) - B
\tag{17}
$$

If the condition in Line 28 of Algorithm 1 is already satisfied, we can just use $x_v = \tilde{x}_v$, which directly satisfies equation 16. Otherwise, we need to follow the expert by setting $x_v = x_v^\pi$. We prove $x_v = x_v^\pi$ satisfies the robustness condition at any step $v$.

From Lemma C.2, since $0 \leq \rho \leq 1$ and $\Delta f_{x_v^\pi} \geq 0$ we have

$$\Delta f_{x_v^\pi} \geq \rho \left( \Delta f_{x_v^\pi}^\pi + G\left(x_v^\pi, \{\mathcal{V}_{u,v-1}\}_{u\in\mathcal{U}}, \{\mathcal{V}_{u,v}^\pi\}_{u\in\mathcal{U}}\right) - \tilde{G}\left(\{\mathcal{V}_{u,v-1}\}_{u\in\mathcal{U}}, \{\mathcal{V}_{u,v-1}^\pi\}_{u\in\mathcal{U}}\right) \right).$$

Then, by substituting it back to Eqn. equation 17, we have

$$\begin{aligned}
R_{v-1} + \Delta f_{x_v^\pi} \geq & \rho \left( \Delta f_{x_v^\pi}^\pi + G\left(x_v^\pi, \{\mathcal{V}_{u,v-1}\}_{u\in\mathcal{U}}, \{\mathcal{V}_{u,v}^\pi\}_{u\in\mathcal{U}}\right) - \tilde{G}\left(\{\mathcal{V}_{u,v-1}\}_{u\in\mathcal{U}}, \{\mathcal{V}_{u,v-1}^\pi\}_{u\in\mathcal{U}}\right) \right) \\
& + \rho \left( R_{v-1}^\pi + \tilde{G}\left(\{\mathcal{V}_{u,v-1}\}_{u\in\mathcal{U}}, \{\mathcal{V}_{u,v-1}^\pi\}_{u\in\mathcal{U}}\right) \right) - B \\
= & \rho \left( R_{v-1}^\pi + \Delta f_{x_v^\pi}^\pi + G\left(x_v^\pi, \{\mathcal{V}_{u,v-1}\}_{u\in\mathcal{U}}, \{\mathcal{V}_{u,v}^\pi\}_{u\in\mathcal{U}}\right) \right) - B \\
= & \rho \left( R_v^\pi + \tilde{G}\left(\{\mathcal{V}_{u,v}\}_{u\in\mathcal{U}}, \{\mathcal{V}_{u,v}^\pi\}_{u\in\mathcal{U}}\right) \right) - B.
\end{aligned} \tag{18}$$

Therefore, after the last step $v$, LOMAR must satisfy

$$R_v \geq \rho \left( R_v^\pi + \tilde{G}\left(\{\mathcal{V}_{u,v}\}_{u\in\mathcal{U}}, \{\mathcal{V}_{u,v}^\pi\}_{u\in\mathcal{U}}\right) \right) - B \geq \rho R_v^\pi - B,$$

where $R_v$ and $R_v^\pi$ are the total rewards of LOMAR and the expert algorithm $\pi$ after the last step $v = |\mathcal{V}|$, respectively. Thus, we complete the proof for the free-disposal setting.

