# OpenReview forum: "Learning for Edge-Weighted Online Bipartite Matching with Robustness Guarantees"
_ICLR.cc/2023/Conference — Submitted to ICLR 2023_

### Official Review · Reviewer_7xYF · 2022-10-24

**Confidence:** 3
**Clarity, Quality, Novelty And Reproducibility:** The paper is of good quality, clarity…
**Correctness:** 3
**Technical Novelty And Significance:** 3
**Empirical Novelty And Significance:** 2
**Recommendation:** 5

**Details Of Ethics Concerns:**

No ethics concerns.

**Strength And Weaknesses:**

Strengths
- Important problem that is at the core of many real-life problems.
- Generally well written.
- The combination of RL and robustification is quite interesting the way it is done here

Weaknesses
- It is not clear how LOMAR compares to other robust algorithms for the problem stemming from the "learning augmented algorithms" literature. Although these algorithms are robust as well, it seems conceivable that LOMAR has an advantage by incorporating the robustification into the learning. It would be nice to see at least experimental comparison to these results and maybe a more extensive discussion on differences and similarities (some of these papers are already cited).
- The literature review and some citations are quite off and give the impression that the authors just searched for some keywords and wrote a couple of sentences on some papers that seemed relevant. As an example, Boyar et al., 2016 is used (i) to support that worst case performance is not necessarily good average-case performance (page 1), (ii) as an example for ML-augmented algorithms (page 2) and (iii) to support the definition of "strict" competitive ratio. This paper is a survey on "advice complexity" and although it is not wrong to cite it in the context of this paper, advice complexity (in general) concerns itself with receiving *accurate* advice and investigating what the least possible amount (and appropriate encoding) of this advice is in order to guarantee optimality. This comes in contrast to ML-augmented algorithms where the advice may be suboptimal but still needs to be incorporated in order to provide improved results in case it is adequate. Also for the definition of competitive ratio I would go to the source and cite one of the papers that introduced the concept, or perhaps a book on online algorithms. Also, I am not familiar with the Gupta and Roughgarden paper but I find it strange that it is cited as an ML-augmented algorithms paper -- since it predates the area.

**Summary Of The Paper:**

The paper presents a reinforcement-learning approach (that the authors call LOMAR) for edge-weighted online bipartite matching which is claimed to have good average but also worst case performance. The approach can handle both the no-free-disposal and the free-disposal setting with the second one allowing for dropping some edges from the matching to include some other ones. The main body of the paper concentrates and no-free-disposal and free disposal is deferred to the appendix.

The presented algorithm (LOMAR) is quite technical and the main novelty is involving a "robustness constraints" which helps decide on whether to follow the expert or the RL decision in each step. The algorithm is then analysed in terms of competitive ratio against an expert online algorithm. Finally, the RL training is incorporated within LOMAR by considering a switching operation based on the aforementioned constraint, and experiments are presented.

**Summary Of The Review:**

Overall this is a reasonably good paper, but I feel that more work needs to be done with respect to placing the results within the context of the literature.

---

> ### Author Response · Authors · 2022-11-14
> **Authors' response to Reviewer 7xYF**
>
> We're happy to address the reviewer's comments.
>
> * **`Comparison with other robustness algorithms stemming from ML-augmented online algorithms`**
>
> We'd like to highlight that ML-augmented online algorithms typically assume a pre-trained black-box ML model. Also, to our knowledge, there does not exist any ML-augmented online algorithm that specifically addresses edge-weighted online bipartite matching in our setting.
>
> Nonetheless, the baseline **DRL-OS** in our experiments, which applies online switching to the pure RL policy trained without awareness of the switching, is actually already a robustness algorithm that falls into the class of ML-augmented algorithms. Specifically, by viewing pure DRL as the black-box ML model, our online switching design (**Section 4**) serves as an analog of the existing ML-augmented algorithms.
>
> Our experimental results (e.g., **Table 1**) show that DRL-OS cannot achieve as high average rewards as LOMAR, which trains the RL policy by explicitly accounting for online switching.
> While it'd be an interesting future direction to explore ML-augmented online algorithms for edge-weighted online bipartite matching, our study highlights a key advantage of incorporating robustification into the training process (which is also pointed out by the reviewer).
>
> * **`Position of our work, and citation of related works`**
>
> We aim at achieving both good average performance and guaranteed worst-case robustness. Specifically, we study an important novel objective --- maximizing the average reward subject to worst-case robustness guarantees --- for edge-weighted online bipartite matching. This is crucial in practice, but has not been studied to our knowledge. Specifically, we differ from the literature as follows.
>
> * **LOMAR vs. online algorithms without ML:** Manually-designed online algorithms focus on improving the worst-case competitiveness under various settings, without addressing the average performance. We consider both average performance and worst-case robustness. Some manually-designed algorithms focus on a *stochastic* setting where the arrival order is random and/or the reward of each online item is independently and identically distributed (i.i.d.) within each problem instance $\mathcal{G}$. By contrast, our settings are significantly different ---
> we only require an unknown distribution for the entire problem instance $\mathcal{G}$ while both the rewards and online arrival order within each instance $\mathcal{G}$ can be adversarial in our problem.
>
> * **LOMAR vs. ML-augmented online algorithms:** ML-augmented algorithms typically assume a pre-trained black-box ML model and still focus on the worst-case performance. Specifically, they aim at achieving bounded robustness when ML prediction is extremely bad, and
> also good competitive ratios when ML prediction is nearly perfect. By contrast, LOMAR bounds the worst-case robustness and uses RL to improve the average performance.
>
> * **LOMAR vs. pure RL:** Pure RL-based solutions only consider the average performance, without worst-case robustness guarantees.
>
> * **LOMAR vs. conservative/constrained RL:** Our problem can be broadly viewed as constrained/conservative episodic Markov decision process (MDP), but the existing literature typically considers *average* or *high-probability* constraints over many episodes (each episode being a problem instance). By contrast, we consider a *hard* constraint on *each episode* to achieve guaranteed worst-case robustness, which requires novel techniques to address (Section 4).
>
> ***We have re-organized the citations and related works per the reviewer's suggestion.***
>
> * **`Definition of competitive ratio`**
>
> Our definition of competitive ratio against a given expert is general and also being used in the literature (e.g., the bi-competitive ratio against two different online algorithms in [1]). Additionally, having a constant additive term $B\geq0$ independent of the input in the competitive ratio is for generality and also common in the literature (e.g., [2]). Note that in our experiments, we set $B=0$ to follow the strict definition.
>
> **References:**
>
> [1] Nicolas Christianson, Tinashe Handina, Adam Wierman, "Chasing Convex Bodies and Functions with Black-Box Advice", COLT 2022.
>
> [2] Antonios Antoniadis, Christian Coester, Marek Eliás, Adam Polak, and Bertrand Simon. Online Metric Algorithms with Untrusted Predictions. In ICML 2020.

---

> > ### Comment · Reviewer_7xYF · 2022-11-25
> > **Response acknowledgment**
> >
> > Thanks for your response.
> >
> > I am still not sure what the difference between “average performance” and “performance when ML predictions are nearly perfect” is supposed to mean. In my understanding average inputs will be well-represented in the training data and ML will provide nearly perfect predictions in that case. I think there is a confusion (maybe on my side?) between the average case taken over all inputs and the average error in the prediction.
> >
> > In any case, I consider my evaluation still adequate.

---

> > > ### Author Response · Authors · 2022-11-25
> > > **Authors' response to follow-up questions**
> > >
> > > We're glad to hear from the reviewer and grateful for the reviewer's appreciation of novelty of our work. We clarify the confusion below.
> > >
> > > - **"performance when ML predictions are nearly perfect"**: This is the worst-case competitive ratio studied by the existing ML-augmented algorithms literature (e.g., [2]) under an *idealized* scenario when the ML predictions are nearly the same as the offline optimal solution (i.e., the competitive reward ratio of pure ML against the offline optimal solution is close to 1). Note that "ML predictions" refer to ML's actions in our context, since the ML model directly outputs online actions, which is also consistent with the existing ML-augmented algorithm designs (e.g., [1]).
> > >
> > > - **"average performance"**: This is the average reward performance $\mathbb{E}_{\mathcal{G}}\left[R(\mathcal{G})\right]$ that we consider in our work. It is true that "average inputs will be well-represented in the training data", which is also the main reason why ML is (often) performing well on average. But, pure ML-based solutions typically do not have worst-case guarantees, and also may not perform well when the training and testing distributions are inconsistent or the ML model is mis-specified (see, e.g., [3] for discussion on a different problem of online allocation). Thus, LOMAR addresses this crucial limitation of pure ML-based solutions by providing worst-case performance guarantees.
> > >
> > > We hope our response clarifies the confusion and welcome any additional feedback the reviewer may have.
> > >
> > > **References:**
> > >
> > > [1] Nicolas Christianson, Tinashe Handina, Adam Wierman, "Chasing Convex Bodies and Functions with Black-Box Advice", COLT 2022.
> > >
> > > [2] Antonios Antoniadis, Christian Coester, Marek Eliás, Adam Polak, and Bertrand Simon. Online Metric Algorithms with Untrusted Predictions. In ICML 2020.
> > >
> > > [3] http://ai.googleblog.com/2022/09/robust-online-allocation-with-dual.html

---

> > > > ### Comment · Reviewer_7xYF · 2022-11-28
> > > > **answer**
> > > >
> > > > I think we are beating around the bush here. The whole point of the existing ML-augmented algorithms literature is that on top of the \emph{consistency} that you describe in your response it does also include "worst-case guarantees" and performs well when "the training and testing distributions are inconsistent or the ML model is mis-specified" via them proving \emph{robustness}. So it is still unclear to me what differentiates your "average performance" from ML-augmented approaches.
> > > >
> > > > With respect to the previous comment, I am of course aware of the definition of the competitive ratio and also that one can set the additive constant to 0 to obtain the so-called \emph{competitive ratio}. My comment was just that you should not cite a random paper that uses the definition. I think it is ok to provide the definition without citation, but if you need to cite something, then it is a good idea to cite either the original paper by Sleator and Tarjan from the 80's or some textbook.

---

> > > > > ### Author Response · Authors · 2022-11-28
> > > > > **Authors' response**
> > > > >
> > > > > We appreciate the reviewer's response and are glad to further clarify the confusion.
> > > > >
> > > > > * **`Average performance`**
> > > > >
> > > > > While it's true that both ML-augmented algorithms and LOMAR can provide robustness guarantees, the way they treat *average performance* is dramatically different. Specifically, ML-augmented algorithms apply a downstream algorithmic step (i.e., modified online algorithm) after an ML model output and typically assume that the ML model is trained as a `standalone` optimizer --- an optimizer that performs well on average on its own. Clearly, there is a mismatch between the ML training process (oblivious of the downstream algorithmic step) and testing process (in the presence of the downstream algorithmic step). Then, ML-augmented algorithms address this "training-testing" mismatch and attempt to preserve the good average performance of the original ML model by offering "consistency". That is, by "consistency", the average performance of the ML model is preserved to the extent as specified by the "consistency" measure (still a worst-case metric). By sharp contrast, LOMAR addresses the "training-testing" match by training the ML model with explicit awareness of the downstream algorithmic step (**Section 5** where we introduce a softmax probability based on Eqn. 4 to approximate the hard switching process during training and enable gradients to propagate back to the ML model parameters). This directly optimizes the *average performance* of the entire algorithm (ML model plus the downstream algorithmic step).
> > > > >
> > > > > Let's also explain this point more formally. Denote the overall algorithm as $ALG=Rob \circ \theta$, where $Rob$ denotes the downstream algorithmic step (i.e., switching operation in LOMAR or specific algorithmic designs in ML-augmented algorithms), $\theta$ represents the ML model parameter, and $\circ$ denotes the algorithm composition. Thus, given an algorithm, the actual average reward is $\mathbb{E}_{\mathcal{G}}\left[R(\mathcal{G})|ALG=Rob\circ\theta\right]$, where $R(\mathcal{G})$ is the reward for instance $\mathcal{G}$.
> > > > >
> > > > > --- In ML-augmented algorithms, the ML model parameter is optimized for $\mathbb{E}_{\mathcal{G}}\left[R(\mathcal{G})|ML=\theta\right]$ as a standalone optimizer without being aware of the downstream algorithmic step $Rob$. That is, the training objective differs from the actual testing objective, creating a mistmatch.
> > > > >
> > > > >
> > > > > --- LOMAR directly optimizes the ML model parameter for the actual average reward $\mathbb{E}_{\mathcal{G}}\left[R(\mathcal{G})|ALG=Rob\circ\theta\right]$ by considering the downstream algorithmic step $Rob$. That is, the training and testing objectives are the same.
> > > > >
> > > > > The average performance degradation due to training-testing objective mismatches has also been recently recognized in an orthogonal emerging area of decision-focused learning (e.g., [1]) where the goal is to train the ML model by explicit considering the downstream process and the ultimate goal. Note that decision-focused learning often assumes a pre-determined downstream process, rather than designing a new algorithmic step to ensure robustness as is being done in LOMAR.
> > > > >
> > > > > Finally, our empirical results (e.g., Table 1) also support our point that the training approach in existing ML-augmented algorithms (i.e., DRL-OS which can be viewed as an ML-augmented algorithm) cannot achieve as high average rewards as LOMAR that trains the RL policy by explicitly accounting for online switching.
> > > > >
> > > > > **In summary,** while both ML-augmented algorithms and LOMAR can provide robustness guarantees, LOMAR trains the ML model directly to optimize for the average reward performance, which significantly differs from the existing ML-augmented approaches (for other problems and not yet available for our problem setting of edge-weighted online bipartite matching).
> > > > >
> > > > >
> > > > > * **`Citation`**
> > > > >
> > > > > Thanks! We'll cite a book for the definition of competitive ratio as per the reviewer's suggestion.
> > > > >
> > > > > **References:**
> > > > >
> > > > > [1] Sanket Shah, Kai Wang, Bryan Wilder, Andrew Perrault, Milind Tambe, "Decision-Focused Learning without Differentiable Optimization: Learning Locally Optimized Decision Losses," https://arxiv.org/abs/2203.16067

---

### Official Review · Reviewer_D8g2 · 2022-10-26

**Confidence:** 5
**Correctness:** 4
**Technical Novelty And Significance:** 3
**Empirical Novelty And Significance:** 3
**Recommendation:** 5

**Clarity, Quality, Novelty And Reproducibility:**

Clarity

+ The paper is clear and well-written. The literature survey and placement is done fairly.


Quality

+ The considered research is high-quality and solving an important problem. Overall the direction is significant.

Novelty

+ The considered robust RL algorithm for Online matching is very novel. These combine two different areas of research that don't often intersect. The paper is novel in combining disparate ideas together.

Reproduciblity

+ As far as I can tell, the paper contains sufficient details to reproduce the experiments

**Strength And Weaknesses:**

Strengths

+ Robustifying RL algorithms are practically important. Online Matching problems are also practically important with many economic consequences. Thus, I find combing Robust RL area with online matching an important problem and understanding the limits, foundations and novel algorithms is a key contribution. This paper makes a step and adds to this direction.

+ The paper is well-written and overall, the ideas are clearly explained. The online switching based algorithm is intuitive and practical.

+ The paper considers an exhaustive space of both non-free disposal and free disposal settings which are both plently available in practical settings such as ads and recommender systems.


Weakness

- The primary weakness I find is in experiments. In particular, the studied problem is actually theoretically solvable using worst-case online algorithms when the inputs are stochastic. I wonder if the authors can compare against these optimal algorithms in the experimental section. In particular, when the inputs are i.i.d., these worst case algortihms can achieve competitive ratio > 0.7 in the worst-case and sometimes much higher on a random instance. Given that in the experiments, the competitive ratio obtained is of the order of 0.7-0.8, I think this comparison would make the paper really strong.




**Summary Of The Paper:**

This paper studies the problem of online weighted b-matching problem augmented with an RL-based expert algorithm. The goal of the RL algorithm is to leverage the expert such that when the RL algorithm's decision is bad while the expert algorithm's decision is good, the RL algorithm achieves robustness by using the expert algorithm's choice, while also having high enough average performance across the board, even when the expert is bad. They study this problem both in the free disposal and no-free disposal setting (where you can't throw away already matched edges) and show empirically competitive results that improves over the RL-only based algorithm.

**Summary Of The Review:**

Overall, I am on the border since I would like the empirical section to be strengthened. In particular, the paper introduces complex ideas and would like to see when this would significantly perform better than hand-tuned optimal online algorithms and when it would perform worse. This kind of extensive analysis will be useful to future research as well as a practitioner who would like to try these ideas in the wild.

---

> ### Author Response · Authors · 2022-11-14
> **Authors' response to Reviewer D8g2**
>
> We're happy to address the reviewer's comments.
>
> * **`Empirical results when our problem is theoretically solvable using worst-case online algorithms when the inputs are stochastic`**
>
> We would like to first draw the `crucial difference` between our setting and the *stochastic setting* considered by the existing worst-case online algorithms. More specifically, as highlighted in our response to all the reviewers, we consider an unknown distribution of problem instances $\mathcal{G}=(\mathcal{U},\mathcal{V},\mathcal{W})$, whereas problem instances with adversarial rewards and/or online arrival orders are still possible. By contrast, the *stochastic input setting* considered by existing worst-case online algorithms refers to inputs in each problem instance and is different [1], e.g., random order (i.e., the rewards can be adversarially chosen but the arrival order is random within each problem instance) and known/unknown i.i.d. (i.e., the reward of each online item is drawn from an i.i.d. distribution within each problem instance).
>
> *`Interpretation of robustness:`*
> The low theoretical competitive ratios (against OPT) for general edge-weighted online bipartite matching is mainly due to the intrinsic difficulty of the problem. Nonetheless, the empirical performance of an expert algorithm can still be good. For example, an empirically strong expert is greedy [2], whose empirical competitive ratio is around 0.7 depending on the graph parameter. In our study, LOMAR can work with any expert algorithm while satisfying the worst-case robustness relative to the given expert. Thus, if we consider an effective algorithm currently used in practice as the expert, LOMAR can be viewed as a robust solution that leverages the benefits of RL (to improve the average performance) while still being able to guarantee $\rho$-competitive against the expert for *any* problem instance. Such competitiveness against an existing online algorithm is also common and being studied in the literature (e.g., [4]).
>
> *`Empirical results:`*  We have shown the results for settings when the expert has good empirical competitive ratios (e.g., 0.723 in Section 6). In this case, pure RL also works very well on average, and hence even by choosing a small $\rho$ (e.g., 0.4), LOMAR can still offer a good average performance as well as an improved competitive ratio (**Table 1**).
>
> *`New results for a new expert (OSM):`* Per the reviewer's suggestion, we add a random order setting in experiments and consider OSM as the expert (Online Secretary Matching, which has the optimal competitive ratio of $1/e$) [3]. Unfortunately, there doesn't exist an online algorithm that can achieve better competitive ratios for our problem setup, even by assuming unknown i.i.d. rewards within each problem instance [1]. Better competitive ratios are only possible by further relaxing the assumptions (e.g., unweighted bipartite matching or allowing each online item to receive items beyond its capacity) [1], which do not apply to our considered setting for experiments.
>
> We show the results for OSM in **Figure 4** and **Table 2** in Appendix A.1.3. In terms of the empirical worst-case cost ratio, OSM doesn't work as well as Greedy (partly because OSM skips the first $|\mathcal{V}|/e$ online items). Nonetheless, LOMAR still achieves good average performance with guaranteed robustness. For example, with $\rho=0.6$, LOMAR achieves both higher average and worst-case rewards than OSM, while theoretically achieving 0.6-competitive against OSM for any instance (which also translates into guaranteed $0.6/e$ competitive against OPT under the random order assumption).
>
> *`Summary:`* In general, $\rho\in[0,1]$ governs the tradeoff between the average performance and the level of guaranteed robustness: the larger $\rho$, the more robustness, and vice versa. Practically, this implies that if the expert works well in the worst case, one can set a smaller $\rho$ to further improve the average performance while achieving good robustness. This is also confirmed by our empirical results. Importantly, LOMAR can work with any expert and is the first to achieve good average performance while guaranteeing the worst-case robustness relative to the given expert.
>
>
> **References:**
>
> [1] Aranyak Mehta. Online matching and ad allocation. Foundations and Trends in Theoretical Computer Science, 8 (4):265–368, 2013. URL http://dx.doi.org/10.1561/0400000057
>
> [2] Mohammad Ali Alomrani, Reza Moravej, and Elias B. Khalil. Deep policies for online bipartite matching: A reinforcement learning approach. CoRR, abs/2109.10380, 2021. URL https://arxiv.org/abs/2109.10380
>
> [3] Thomas Kesselheim, Klaus Radke, Andreas Tönnis, and Berthold Vöcking. An optimal online algorithm for weighted bipartite matching and extensions to combinatorial auctions. In European symposium on algorithms, 2013.
>
> [4] Nicolas Christianson, Tinashe Handina, Adam Wierman, "Chasing Convex Bodies and Functions with Black-Box Advice", COLT 2022.

---

### Official Review · Reviewer_XizC · 2022-10-26

**Confidence:** 3
**Correctness:** 4
**Technical Novelty And Significance:** 2
**Empirical Novelty And Significance:** 2
**Recommendation:** 3

**Clarity, Quality, Novelty And Reproducibility:**

I found the paper a bit hard to read and follow - as I mentioned earlier in the review, in my view, the paper focused on the wrong aspects of the work. The switching framework itself is not particularly novel within the online algorithms and is indeed the first thing one would think of when faced with such a question.

**Strength And Weaknesses:**

Strengths:
+ Unlike most work on learning augmented algorithms that aim to augment online algorithms with some predictions / advice from ML models; the paper takes the complementary view of an online algorithm used as a safety railing to guide a ML model.

Weaknesses:
- The model feels too brittle - in the general case when the edge weights are unknown - the algorithm essentially only follows the decisions of the expert and can not use the RL agent at all. Maybe a restriction to a more specific problem would yield better insights.

- It’s not clear what the focus of the paper is. Section 4 is concerned with how to use the RL agent’s decisions while maintaining robustness against a fixed expert. It would be better to restructure this as a meta-algorithm that takes in two algorithms A (expert) and B (RL agent): The meta algorithm switches between the actions of A and B and guarantees to maintain a given robustness wrt A while maximizing the number of times it uses actions of B. (Note that this section is completely independent of RL). Indeed, viewed in this light, the meta-algorithm and its analysis is almost trivial. [ Also worth referring to “combining” algorithms for paging and metrical task systems (e.g. Fiat et al)].

- Section 5 then deals with the challenges of training the RL agent when used along with the switching algorithm above. I would much rather see more time and space allotted to this section and clarify the training process in more details.

- The empirical section includes preliminary experiments and does not demonstrate strong positive results.


**Summary Of The Paper:**

The paper proposes a RL-based approach for online weighted bipartite matching. A key novelty of the paper is to augment the decisions of the RL agent with a classic online algorithm to obtain robustness guarantees. Upon the arrival of an online vertex, the algorithm queries both a base online algorithm (called expert) and the RL agent. It follows the action of the RL agent unless doing so has the possibility that the algorithm incurs significant loss with respect to the expert.

**Summary Of The Review:**

The paper proposes a framework to use an expert algorithm to guide the decisions of an RL agent for online bipartite matching. I like the problem direction but find the current paper version a bit lacking with respect to details of (i) RL training procedure, (ii) Difficulty in naively adapting standard RL techniques, (iii) thorough empirical evaluation.

---

> ### Author Response · Authors · 2022-11-14
> **Authors' response to Reviewer XizC (Part 2)**
>
> * **`Empirical performance`**
>
> Our goal is *not* to achieve the best average performance and worst-case performance *simultaneously*, which is impossible in general settings; our goal is *not* to achieve the best average performance or worst-case robustness *alone* either, which have been separately studied by pure RL algorithms and pure manually-designed algorithms, respectively. Instead, we aim at achieving both good average performance and guaranteed worst-case robustness, which is an important and novel objective in practice. Also, by setting $\rho=1$ and $\rho=0$, LOMAR can reduce to the expert and pure RL, respectively. Thus, our study generalizes the existing online algorithms that are either focusing on the average performance or the worst-case performance.
>
> Thus, we respectfully disagree with the reviewer's point that our empirical evaluation "does not demonstrate strong positive results". In fact, our empirical results in Section 6 and Appendix A adequately demonstrate the key merit of our algorithm: Good average performance with guaranteed worst-case robustness. Moreover, when there's a training-testing distributional shift, pure RL can perform arbitrarily badly, while LOMAR still has performance guarantees. Note finally that the seemingly small differences in terms of the absolute reward values are actually due to the problem settings and common in the literature [1].
>
> **References:**
>
> [1] Mohammad Ali Alomrani, Reza Moravej, and Elias B. Khalil. Deep policies for online bipartite matching: A reinforcement learning approach. CoRR, abs/2109.10380, 2021. URL https://arxiv.org/abs/2109.10380

---

> ### Author Response · Authors · 2022-11-14
> **Authors' response to Reviewer XizC (Part 1)**
>
> We’re happy to address the reviewer’s comments.
>
> * **`Known weights`**
>
> We focus on edge-weighted online bipartite matching with *known* weights revealed online to the agent. This is an important and standard setting that has been widely studied in online bipartite matching [1,2] and has many practical motivations [1]. The setting of completely *unknown* weights is orthogonal to our study.
>
> * **`Focus of our study and switching condition`**
>
> We aim at achieving both good average performance and guaranteed worst-case robustness. Specifically, we study a novel objective --- maximizing the average reward subject to worst-case robustness guarantees --- for edge-weighted online bipartite matching. This is an important goal in practice but, to our best knowledge, has not been well studied before (see *Differences from the literature* in our response to all the reviewers for detailed comparison).
>
> We respectfully disagree with the reviewer's point that our *switching condition* is trivial. While switching is common in online algorithms, **"how to switch"** (i.e. the switching condition) is highly non-trivial and a key merit for algorithm designs [3,4,5]. For example, the study [3] (as pointed by the reviewer) switches between different online experts based on a predetermined threshold, and the optimal cost ratio is 9 for two experts (i.e., 9 times of the robust expert’s cost) as shown in a recent study [5]. Considering smoothed online optimization, the authors in [4] design a customized switching condition to further improve the cost ratios. The study [6] switches between a non-conservative policy and a conservative policy for MDP based on their value functions without the worst-case robustness guarantee.
>
> By contrast, we consider a different problem --- online bipartite matching --- and propose a novel reservation reward (i.e., `Eqn. 3 for no-free-disposal and Eqn. 7 for free-disposal`) for the switching condition. Compared to the existing switching conditions [3,4,5], our novelty is the new carefully-crafted reservation reward that hedges against the expert's future possible reward advantages, thus achieving $R(\mathcal{G})\geq \rho R^{\pi}(\mathcal{G})-B$ for any problem instance $\mathcal{G}$ and **any $\rho\in[0,1]$** and $B\geq0$.
>
> Therefore, the switching condition in Section 4 is novel and, to our knowledge, has not been considered for achieving worst-case robustness.
>
> * **`RL training in Section 5`**
>
> We train an RL agent to improve the average performance subject to the worst-case robustness constraint. This is different from the existing ML-augmented algorithms (e.g., [4]) that assume a pre-trained black-box ML model and still focus on the worst-case performance.
> Also, due to the novel switching condition, the RL training (presented in Algorithm 3 in the appendix) differs from the standard process. Specifically, we introduce a softmax probability based on Eqn. 4 to approximate the hard switching process during training. This softmax probability essentially enables gradients to propagate back to the RL model parameters. At the beginning of RL training, we can set a high temperature to encourage more aggressive exploration without being stuck to the expert's policy; as the RL model performance continuously improves, we can reduce the temperature in order to make the RL agent more aware of the downstream switching operation.
>
> **References:**
>
> [1] Mohammad Ali Alomrani, Reza Moravej, and Elias B. Khalil. Deep policies for online bipartite matching: A reinforcement learning approach. CoRR, abs/2109.10380, 2021. URL https://arxiv.org/abs/2109.10380
>
> [2] Aranyak Mehta. Online matching and ad allocation. Foundations and Trends in Theoretical Computer Science, 8 (4):265–368, 2013. URL http://dx.doi.org/10.1561/0400000057
>
> [3] Fiat, A., Rabani, Y., and Ravid, Y. Competitive k-server algorithms. J. Comput. Syst. Sci., 48(3):410–428, 1994.
>
> [4] Daan Rutten, Nico Christianson, Debankur Mukherjee, and Adam Wierman. Online optimization
> with untrusted predictions. CoRR, abs/2202.03519, 2022.
>
> [5] Antonios Antoniadis, Christian Coester, Marek Eliás, Adam Polak, and Bertrand Simon. Online Metric Algorithms
> with Untrusted Predictions. In ICML 2020.
>
> [6] Yunchang Yang, Tianhao Wu, Han Zhong, Evrard Garcelon, Matteo Pirotta, Alessandro Lazaric, Liwei Wang, and Simon Shaolei Du. A reduction-based framework for conservative bandits and reinforcement learning. In ICLR, 2022.

---

### Author Response · Authors · 2022-11-14
**General responses to all the reviewers**

We appreciate all the reviewers' comments. We'd first like to highlight our objective and how it differs from the literature as follows.

**`Objective:`**

We denote a problem instance as $\mathcal{G}=\left(\mathcal{U},\mathcal{V},\mathcal{W}\right)$, where $\mathcal{U}$, $\mathcal{V}$, and $\mathcal{W}$ represent the offline items, online items, and edge weights (i.e., rewards), respectively. Given an expert algorithm $\pi$, our goal is to maximize the average reward subject to the worst-case robustness constraint (specified by the hyperparameters $\rho\in[0,1]$ and $B\geq0$):

$\max \mathbb{E}_{\mathcal{G}}\left[R(\mathcal{G})\right],\text{ subject to }R(\mathcal{G})\geq \rho R^{\pi}(\mathcal{G})-B,\forall \mathcal{G},$

where $R(\mathcal{G})$ is the reward for problem instance $\mathcal{G}$ and the expectation $\mathbb{E}_{\mathcal{G}}\left[R(\mathcal{G})\right]$ is over the randomness of the problem instance $\mathcal{G}$. Importantly, we consider an unknown distribution of $\mathcal{G}$, whereas the rewards of different online items and the arrival order within each instance $\mathcal{G}$ can be adversarial. Note that the average reward is complementary to but differs from the worst-case reward, and hence the *`per-instance`* reward constraint in our problem is not redundant. Also, both average and worst-case performances are important in practice (see [1] for a discussion on online resource allocation).


**`Differences from the literature:`**

* **LOMAR vs. online algorithms without ML:** Some manually-designed online algorithms focus on “adversarial” settings with the goal to guarantee the worst-case competitiveness, without addressing the average performance. We consider both average performance and worst-case robustness. Some manually-designed algorithms focus on *stochastic* settings where the arrival order is random and/or the reward of each online item is independently and identically distributed (i.i.d.) within each problem instance $\mathcal{G}$. By contrast, our settings are significantly different ---
we only require an unknown distribution for the problem instance $\mathcal{G}$, while problem instances with adversarial rewards and/or online arrival orders are still possible.

* **LOMAR vs. ML-augmented online algorithms:** ML-augmented algorithms typically assume a pre-trained black-box ML model and still focus on the worst-case performance. Specifically, they aim at achieving bounded robustness when ML prediction is extremely bad, and
also bounded competitive ratios when ML prediction is nearly perfect. By contrast, LOMAR bounds the worst-case robustness and uses RL to improve the average performance.

* **LOMAR vs. pure RL:** Pure RL-based solutions only consider the average performance, without worst-case robustness guarantees.

* **LOMAR vs. conservative/constrained RL:** Our problem can be broadly viewed as constrained/conservative episodic Markov decision process (MDP), but the existing literature typically considers *average* or *high-probability* constraints over many episodes (each episode being a problem instance). By contrast, we consider a *hard* constraint on *each episode* to achieve guaranteed worst-case robustness, which requires novel techniques to address (Section 4).

**`Summary of contributions:`**

Our goal is *not* to achieve the best average performance and worst-case robustness *simultaneously*, which is impossible in our general settings; our goal is *not* to achieve the best average performance or worst-case robustness *alone* either, which have been separately studied by pure RL algorithms and pure manually-designed algorithms, respectively. Instead, we aim at achieving both good average performance and guaranteed worst-case robustness. Specifically, we study an important novel objective --- maximizing the average reward subject to worst-case robustness guarantees --- for edge-weighted online bipartite matching. We focus on a setting where the problem instance $\mathcal{G}$ follows an unknown distribution, whereas both the adversarial instance $\mathcal{G}$ in terms of online arrival order and rewards can exist. To achieve the worst-case robustness constraint against a given expert $\pi$, we introduce a novel reservation term and design a new switching condition (**Section 4**); to improve the average reward, we propose new RL training by explicitly taking into account the novel switching condition (**Section 5**). Finally, we run experiments to support our analysis.

We have also updated the paper to better highlight our contributions.


**References:**

[1] https://ai.googleblog.com/2022/09/robust-online-allocation-with-dual.html

---

### Decision · Program_Chairs · 2023-01-20

**Decision:**

Reject

**Justification For Why Not Higher Score:**

The scope is a bit outside of ICLR. If you want, you can promote it. I would be personally totally fine with it.

**Justification For Why Not Lower Score:**

N/A

**Metareview: Summary, Strengths And Weaknesses:**

This paper looks at a variant of online bipartite matching. This is a neat paper... but the major concern that I have is its relevancy to the ICLR community.

As a consequence, I asked several reviewers about their opinion which is unfortunately a bit negative. I take from this that ICLR is not the best fit for such a paper.

Had it been in another conference, certainly more TCS than ML, it would have been accepted (almost for sure). I am not certain about the motivations behind the choice of submission at ICLR, but I would happily accept this paper in a different venue if I am handling it again in the future. I am sorry to give positive feedback and then to reject a paper, but I think finding the appropriate conference/journal is important.